# A normalised framework for the Zero Emissions Commitment

Richard G. Williams<sup>1</sup>, Philip Goodwin<sup>2</sup>, Paulo Ceppi<sup>3</sup>, Chris D. Jones<sup>4,5</sup>, and Andrew MacDougall<sup>6</sup>

<sup>1</sup>Department of Earth, Ocean, and Ecological Sciences, School of Environmental Sciences, University of Liverpool, Liverpool, United Kingdom

**Correspondence:** Richard G. Williams (ric@liverpool.ac.uk)

Abstract. The Zero Emissions Commitment (ZEC) measures the transient climate response after carbon emissions cease, defined by whether there is a continued rise or decrease in global surface temperature. This delayed climate response affects the maximum cumulative carbon emission to avoid exceeding a warming target. In a set of 9 Earth system models following an idealised atmospheric CO<sub>2</sub> scenario with a cumulative emission of 1000 PgC, the ZEC after 50 years ranges from  $-0.3^{\circ}$ C to 0.28°C with a model mean of -0.11°C and standard deviation of 0.19°C. In order to understand these different climate responses, a geometric measure of the ZEC is introduced, measuring the fraction of warming relative to the time of zero emissions, and connected via a normalised framework to carbon, radiative and thermal processes. Inter-model differences in the ZEC are primarily due to differences in the radiative response, planetary heat uptake and the land carbon sink, with more minor contributions from differences in the ocean carbon sink and climate feedback. The ZEC response is controlled by opposing-signed contributions: (i) cooling from a decrease in radiative forcing from a carbon contribution due to increasing land and ocean carbon uptake, versus (ii) surface warming from a thermal contribution involving a decline in the fraction of radiative forcing used for planetary heat uptake plus possible amplification by climate feedback. The carbon contribution to the ZEC depends on the increase in the ocean carbon sink and whether the land carbon sink either increases or saturates in time. The thermal contribution to the ZEC depends upon how radiative forcing is partitioned between planetary heat uptake and radiative response with the radiative response either declining in time or remaining constant. These inferences as to the controls of the ZEC broadly carry over for diagnostics for a large ensemble, observationally-constrained, efficient Earth system model using two different emission scenarios to reach net zero. The large set of ensembles reveal a partial compensation between the changes in landborne and oceanborne fractions, as well as including ensembles with a greater range in amplification of warming by climate feedbacks.

#### 20 1 Introduction

Climate models reveal a near-linear dependence of the global surface temperature change with cumulative carbon emissions in experiments following idealised CO<sub>2</sub> experiments (Matthews et al., 2009; Allen et al., 2009; Zickfeld et al., 2009; Gillett

<sup>&</sup>lt;sup>2</sup>School of Ocean and Earth Sciences, University of Southampton, Southampton, UK

<sup>&</sup>lt;sup>3</sup>Department of Physics, Imperial College London, London, UK

<sup>&</sup>lt;sup>4</sup>Hadley Centre, UK Met Office, Exeter, UK

<sup>&</sup>lt;sup>5</sup>School of Geographical Sciences, University of Bristol, Bristol, UK

<sup>&</sup>lt;sup>6</sup>St. Francis Xavier University, Nova Scotia, Canada

et al., 2013; Collins et al., 2013). Once carbon emissions cease, climate models suggest either a slight increase or decrease in surface temperature. This delayed climate response to past carbon emissions is important for policy makers as this response affects the maximum amount of carbon that may be emitted before exceeding a warming target (Allen et al., 2022; Matthews and Zickfeld, 2012).

These different phases of the climate response to carbon emissions are represented by two climate metrics. The first climate metric relevant during emissions, the Transient Climate Response to Cumulative CO<sub>2</sub> Emissions (TCRE), measures the dependence of surface warming to cumulative carbon emissions (Matthews et al., 2009; Gillett et al., 2013; MacDougall, 2016; Williams et al., 2016; Matthews et al., 2018; Jones and Friedlingstein, 2020; Williams et al., 2020). Individual climate models reveal a nearly constant value for the TCRE over a centennial timescale, although the value of the TCRE varies between individual climate models (Gillett et al., 2013; Williams et al., 2017). The second climate metric relevant after emissions cease, the Zero Emissions Commitment (ZEC), measures the temperature change after the time of net zero and represents the warming that might be in the pipeline from past emissions. Most climate models reveal a slight cooling with a negative ZEC, although some individual models reveal a slight warming with a positive ZEC (MacDougall et al., 2020). This ZEC response may also be modified by the emission scenario (Sanderson et al., 2024). There is significant uncertainty as to which processes control the ZEC and different processes dominate according to the timescale of interest, ranging from decades to millennia; see the review by Palazzo Corner et al. (2023).

Our aim is to understand the competing effects of thermal, radiative and carbon processes in controlling the climate response post emissions as represented by the ZEC. A geometric measure of the ZEC is introduced, measuring the fraction of warming relative to the time of zero emissions. The geometric ZEC is connected via a normalised framework to changes in thermal, radiative and carbon contributions (Section 2). Each of these contributions are connected and interpreted in terms of the empirical energy balance at the top of the atmosphere, the dependence of the radiative forcing on atmospheric CO<sub>2</sub> and the global carbon inventory. This framework is applied to diagnostics for a suite of Earth system models following the Zero Emissions Commitment Model-Intercomparison Project (ZECMIP) (Jones et al., 2019; MacDougall et al., 2020), involving a 1% annual rise in atmospheric CO<sub>2</sub> (referred to as 1pctCO<sub>2</sub>) until a particular cumulative carbon emission is reached and then emissions cease (Section 3). These diagnostics are repeated for a large ensemble of observationally-constrained model projections (Goodwin, 2018; Goodwin et al., 2020) following two different choices of emission scenarios, either the same annual rise in atmospheric CO<sub>2</sub> or a constant carbon emission (referred to as flat10 (Sanderson et al., 2024)) until a maximum cumulative carbon emission is reached (Section 4). Finally, the wider implications of the study are discussed and summarised (Section 5).

## 2 Theory

Theoretical identities are set out for the two key climate metrics, the TCRE and ZEC, defining the climate response during emissions and post emissions respectively. The TCRE relationship draws upon prior work, but the application to the ZEC has not been set out before.

### 55 2.1 Identity for the TCRE

The TCRE measures the dependence of surface warming to cumulative  $CO_2$  emissions and is defined by the change in globalmean, surface air temperature,  $\Delta T(t)$  in K, relative to the pre industrial divided by the cumulative carbon emission,  $I_{em}(t)$  in EgC, such that

$$TCRE \equiv \frac{\Delta T(t)}{I_{em}(t)},\tag{1}$$

where  $\Delta$  represents the change since the time of the pre industrial. The TCRE is approximately scenario independent and depends only on the cumulative carbon emissions.

The TCRE from (1) may be related to an identity involving the product of two terms, the Transient Climate Response (TCR) affected by climate processes and the airborne fraction affected by the carbon cycle (Matthews et al., 2009; Solomon et al., 2009; Gillett et al., 2013; MacDougall, 2016; Jones and Friedlingstein, 2020), such that

65 TCRE = 
$$\frac{\Delta T(t)}{I_{em}(t)} = \underbrace{\frac{\Delta T(t)}{\Delta I_A(t)}}_{\text{TCR}} \underbrace{\frac{\Delta I_A(t)}{I_{em}(t)}}_{\text{Iem}(t)},$$
 (2)

where the TCR is defined by the ratio of the surface temperature change,  $\Delta T(t)$ , and change in the atmospheric carbon inventory,  $\Delta I_A(t)$ , and the airborne fraction is defined by the ratio of the change in the atmospheric carbon inventory,  $\Delta I_A(t)$ , and the cumulative carbon emissions,  $I_{em}(t)$ .

The TCRE can also be equivalently defined by separating the TCR term in (2) into a product of two terms, involving separate thermal and radiative dependencies, such that

$$TCRE = \frac{\Delta T(t)}{I_{em}(t)} = \underbrace{\frac{\Delta T(t)}{\Delta F(t)}}_{\text{Lem}(t)} \underbrace{\frac{\Delta F(t)}{\Delta I_A(t)}}_{\text{Lem}(t)} \underbrace{\frac{\Delta I_A(t)}{I_{em}(t)}}_{\text{Lem}(t)},$$
(3)

where the thermal dependence is given by the ratio of the surface temperature change  $\Delta T(t)$  and the change in the radiative forcing,  $\Delta F(t)$ , and the radiative dependence from the ratio of the change in the radiative forcing,  $\Delta F(t)$ , and the change in the atmospheric carbon inventory,  $\Delta I_A(t)$  (Goodwin et al., 2015; Ehlert et al., 2017; Williams et al., 2016, 2017; Katavouta et al., 2018). The benefit of this additional step is to gain insight into the thermal and radiative effects on the TCRE, by drawing upon the energy balance at the top of the atmosphere and the logarithmic dependence of radiative forcing on atmospheric CO<sub>2</sub>.

#### 2.2 Identities for the ZEC

The ZEC measures the temperature change relative to the pre industrial,  $\Delta T(t)$ , minus the temperature change at the time of net zero,  $t_{ZE}$ ,  $\Delta T(t_{ZE})$ , and is defined by

$$ZEC \equiv \Delta T(t) - \Delta T(t_{ZE}). \tag{4}$$

This definition of the ZEC measures the absolute value of the temperature change and is likely to be sensitive to the warming level experienced from the emission scenario.

Alternatively, a geometric measure of the ZEC is given by the ratio of the temperature change,  $\Delta T(t)$ , and the temperature change at the time of net zero,  $\Delta T(t_{ZE})$ , and measures the fractional zero emission commitment,

$$\frac{\Delta T(t)}{\Delta T(t_{ZE})}. (5)$$

A positive ZEC corresponds to this geometric measure,  $\Delta T(t)/\Delta T(t_{ZE}) > 1$ , and a negative ZEC to  $\Delta T(t)/\Delta T(t_{ZE}) < 1$ .

The temperature change,  $\Delta T(t)$ , used to define the ZEC may be related to the cumulative carbon emission,  $I_{em}(t)$ , by the product of the thermal, radiative and carbon-cycle contributions,

$$\Delta T(t) = \frac{\Delta T(t)}{\Delta F(t)} \frac{\Delta F(t)}{\Delta I_A(t)} \frac{\Delta I_A(t)}{I_{em}(t)} I_{em}(t), \tag{6}$$

so that the geometric ZEC from  $\Delta T(t)/\Delta T(t_{ZE})$  may be expressed as a product of normalised thermal, radiative and carbon-cycle contributions,

$$\frac{\Delta T(t)}{\Delta T(t_{ZE})} = \underbrace{\left(\frac{\Delta T(t)}{\Delta F(t)} / \frac{\Delta T(t_{ZE})}{\Delta F(t_{ZE})}\right)}_{\text{thermal}} \underbrace{\left(\frac{\Delta F(t)}{\Delta I_A(t)} / \frac{\Delta F(t_{ZE})}{\Delta I_A(t_{ZE})}\right)}_{\text{carbon cycle}} \underbrace{\left(\frac{\Delta I_A(t)}{\Delta I_A(t_{ZE})}\right)}_{\text{carbon cycle}}.$$
 (7)

The dependence of the emissions is removed after the time of net zero as  $I_{em}(t)$  is taken to be fixed for  $t > t_{ZE}$ .

Our aim is gain insight into the controls of the ZEC and interpret the continued warming or cooling response in terms of its normalised thermal, radiative and carbon-cycle contributions. Next consider the thermal, radiative and carbon-cycle terms in (7) that determine the ZEC response.

## 100 2.2.1 Thermal contribution

95

The thermal contribution may be understood in terms of the energy balance at the top of the atmosphere, where the planetary heat flux into the climate system,  $\Delta N$ , balances the sum of the radiative forcing into the climate system,  $\Delta F$ , and the radiative response,  $\Delta R$  (Gregory et al., 2004; Knutti and Hegerl, 2008; Andrews et al., 2012; Forster et al., 2013),

$$\Delta N(t) = \Delta F(t) + \Delta R(t),\tag{8}$$

where  $\Delta F(t)$  and  $\Delta R(t)$  are defined as positive when supplying energy into the climate system.

The radiative response is parameterised in terms of the product of the climate feedback parameter,  $\lambda(t)$ , and the change in global mean, surface air temperature,  $\Delta T(t)$ ,

$$\Delta N(t) = \Delta F(t) + \lambda(t)\Delta T(t). \tag{9}$$

The dependence of surface temperature on radiative forcing,  $\Delta T(t)/\Delta F(t)$ , in (7) is then directly connected from (9) to the product of the inverse of the climate feedback,  $\lambda(t)^{-1}$ , and the planetary heat uptake divided by the radiative forcing,

 $\Delta N(t)/\Delta F(t)$ ,

$$\frac{\Delta T(t)}{\Delta F(t)} \equiv \frac{1}{\lambda(t)} \frac{\Delta R(t)}{\Delta F(t)} = -\frac{1}{\lambda(t)} \left( 1 - \frac{\Delta N(t)}{\Delta F(t)} \right),\tag{10}$$

where  $(1 - \Delta N(t)/\Delta F(t))$  represents the fraction of the radiative forcing that escapes back to space, rather than being used for planetary heat uptake.

#### 115 2.2.2 Radiative contribution

The radiative forcing,  $\Delta F(t)$ , may be separated into a CO<sub>2</sub> radiative forcing contribution,  $\Delta F_{CO2}(t)$ , and a non-CO<sub>2</sub> radiative forcing contribution,  $\Delta F_{nonCO2}(t)$ , including the contribution of other greenhouse gases and aerosols,

$$\Delta F(t) = \Delta F_{CO2}(t) + \Delta F_{nonCO2}(t)$$

$$= \Delta F_{CO2}(t) \left(1 + \Delta F_{nonCO2}(t) / \Delta F_{CO2}(t)\right). \tag{11}$$

The  $CO_2$  radiative forcing contribution,  $\Delta F_{CO2}(t)$ , may be related to the change in the logarithm of atmospheric  $CO_2$  relative to the pre industrial,

$$\Delta F_{CO2}(t) = a \Delta \ln CO_2(t) = a \left( \ln CO_2(t) - \ln CO_2(t_o) \right), \tag{12}$$

where a is a radiative forcing coefficient in W m<sup>-2</sup> (that is model dependent) and  $t_o$  is the time of the pre-industrial. The change in the logarithm is equivalent to the fractional change,  $\delta \ln x = \delta x/x$ , so that (12) may be written as

125 
$$\Delta F_{CO2}(t) = a \frac{\Delta CO_2(t)}{CO_2(t)} \equiv a \frac{\Delta I_A(t)}{I_A(t)}, \tag{13}$$

where  $I_A(t)$  is the atmospheric inventory of carbon dioxide (defined by the product of the molar mass of the atmosphere and the mixing ratio of atmospheric CO<sub>2</sub>).

The ratio of the change in the radiative forcing from atmospheric CO<sub>2</sub> and atmospheric carbon is then given from (13) by

$$\frac{\Delta F_{CO2}(t)}{\Delta I_A(t)} = \frac{a}{I_A(t)},\tag{14}$$

and the normalised radiative contribution from  $CO_2$  to the ZEC in (7) is given by

$$\frac{\Delta F_{CO2}(t)}{\Delta I_A(t)} / \frac{\Delta F_{CO2}(t_{ZE})}{\Delta I_A(t_{ZE})} = \frac{I_A(t_{ZE})}{I_A(t)}.$$
(15)

and if there are non-CO<sub>2</sub> radiative contributions, then the normalised radiative contribution to the ZEC is then

$$\frac{\Delta F(t)}{\Delta I_A(t)} / \frac{\Delta F(t_{ZE})}{\Delta I_A(t_{ZE})} = \frac{I_A(t_{ZE})}{I_A(t)} \frac{(1 + \Delta F_{nonCO2}(t)/\Delta F_{CO2}(t))}{(1 + \Delta F_{nonCO2}(t_{ZE})/\Delta F_{CO2}(t_{ZE}))}.$$
(16)

#### 2.2.3 Carbon-cycle contribution

The change in atmospheric carbon inventory,  $\Delta I_A(t)$ , is related to the carbon budget involving the cumulative carbon emission,  $I_{em}(t)$ , and the changes in the land and ocean carbon inventories,  $\Delta I_L(t)$  and  $\Delta I_O(t)$ ,

$$\Delta I_A(t) = I_{em}(t) - \Delta I_L(t) - \Delta I_O(t). \tag{17}$$

This response can be expressed in terms of the airborne fraction,  $\Delta I_A(t)/I_{em}(t)$ , varying with the landborne and oceanborne fractions,  $\Delta I_L(t)/I_{em}(t)$  and  $\Delta I_O(t)/I_{em}(t)$  respectively (Jones et al., 2013),

$$\frac{\Delta I_A(t)}{I_{em}(t)} = 1 - \frac{\Delta I_L(t)}{I_{em}(t)} - \frac{\Delta I_O(t)}{I_{em}(t)}.$$
 (18)

## 2.2.4 Mechanistic insight from the ZEC identity

The ZEC response may be affected by a wide range of thermal, radiative and carbon processes, so that isolating their causal effect and comparing their relative importance is challenging to achieve. The benefit of the geometric ZEC and the normalised framework is that there is a more direct link to the thermal, radiative and carbon processes, which is achieved by utilising the top of the atmosphere energy balance (9) and the radiative dependence (13), so that combining (7) with (10) and (16) leads to

$$\frac{\Delta T(t)}{\Delta T(t_{ZE})} = \underbrace{\left(\frac{\lambda(t_{ZE})}{\lambda(t)} \left(1 - \frac{\Delta N(t)}{\Delta F(t)}\right) / \left(1 - \frac{\Delta N(t_{ZE})}{\Delta F(t_{ZE})}\right)\right)}_{\text{thermal}} \times \underbrace{\left(\frac{I_A(t_{ZE})}{I_A(t)} \frac{(1 + \Delta F_{nonCO2}(t) / \Delta F_{CO2}(t))}{(1 + \Delta F_{nonCO2}(t_{ZE}) / \Delta F_{CO2}(t_{ZE}))}\right) \left(\frac{\Delta I_A(t)}{\Delta I_A(t_{ZE})}\right)}_{\text{radiative}}$$
radiative carbon cycle

Hence, whether there is continued warming, a positive ZEC and  $\Delta T(t)/\Delta T(t_{ZE}) > 1$ , or cooling, a negative ZEC and  $\Delta T(t)/\Delta T(t_{ZE}) < 1$ , depends on the time evolution of the products of:

- (i) the thermal contribution involving the climate feedback,  $\lambda(t)$ , and the dependence of the planetary heat uptake on the radiative forcing,  $\Delta N(t)/\Delta F(t)$ ;
- (ii) the radiative contribution involving the atmospheric carbon inventory,  $I_A(t)$ , and the ratio of non-CO<sub>2</sub> radiative forcing and CO<sub>2</sub> radiative forcing,  $\Delta F_{nonCO2}(t)/\Delta F_{CO2}(t)$ ;
  - and (iii) the change in atmospheric carbon,  $\Delta I_A(t)$ , which via the carbon budget (17) is related to the cumulative carbon emissions,  $I_{em}(t)$ , minus the increase in land and ocean carbon inventories,  $\Delta I_L(t) + \Delta I_O(t)$ .

For idealised experiments with only forcing from atmospheric CO<sub>2</sub>, the normalised framework connecting to the geomet-60 ric ZEC (19) simplifies with the normalised radiative contribution given by the ratio of the atmospheric carbon inventory,  $I_A(t_{ZE})/I_A(t)$ , so that

$$\frac{\Delta T(t)}{\Delta T(t_{ZE})} = \underbrace{\left(\frac{\lambda(t_{ZE})}{\lambda(t)}\left(1 - \frac{\Delta N(t)}{\Delta F(t)}\right) / \left(1 - \frac{\Delta N(t_{ZE})}{\Delta F(t_{ZE})}\right)\right)}_{\text{thermal}}\underbrace{\left(\frac{I_A(t_{ZE})}{I_A(t)}\right)}_{\text{radiative}}\underbrace{\left(\frac{\Delta I_A(t)}{\Delta I_A(t_{ZE})}\right)}_{\text{carbon cycle}}.$$

This relationship for the geometric ZEC,  $\Delta T(t)/\Delta T(t_{ZE})$ , can be used to (i) provide mechanistic insight as to the drivers of the temperature change after net zero and (ii) explain inter-model differences in the response of Earth system models after net zero.

Figure 1. Diagnostics of the climate response for a 1pctCO2 experiment with an annual 1% increase in atmospheric CO<sub>2</sub> until the cumulative carbon emission reaches 1000 PgC from ZECMIP (Jones et al., 2019): (a) cumulative carbon emission,  $I_{em}(t)$  in PgC, versus time in years; (b) change in global-mean surface air temperature relative to the pre industrial,  $\Delta T(t)$  in K, versus time; and (c) the change in surface air temperature versus cumulative carbon emissions. The two key climate metrics are defined by these relationships, the TCRE defined by the slope in (c) and the ZEC defined by the temperature change in (b) relative to the time of net zero or by the vertical excursions in (c) after the maximum cumulative carbon emission is reached. The plot includes smoothing of temperature with a 10 year running mean.

#### 2.3 Analyses of ZECMIP responses

#### 2.4 Core experiments

The responses of 9 full Earth system models are analysed following the ZECMIP protocols (Jones et al., 2019; MacDougall et al., 2020), involving an annual 1% rise in atmospheric CO<sub>2</sub> until a cumulative carbon emission of 1000 PgC is reached and then there is no further carbon emission (Fig. 1a). A single realisation is analysed for each model.

Under the ZECMIP protocol, each individual model experiment branches at the time of net zero, one branch continuing with the 1% rise in atmospheric  $CO_2$  and the other branch continuing with no further emissions. The time of net zero as defined by the branch point varies from 61 to 71 years across the set of Earth system models (MacDougall et al., 2020).

Prior to net zero, the global-mean surface temperature increases nearly linearly with the rise in cumulative carbon emissions (Fig. 1b). There is a nearly constant slope of the temperature change versus cumulative carbon emissions up until the maximum emission, which defines the climate metric, the TCRE (Fig. 1c).

After net zero, there are a range of temperature responses from a slight cooling to a slight continued warming (Fig. 1b), where the temperature change relative to the temperature at net zero defines the ZEC. The continued temperature change is also evident in the positive and negative excursions in temperature at the maximum carbon emissions in Fig. 1c.

The temperature response after net zero is interpreted in terms of a geometric ZEC involving the continuing temperature change,  $\Delta T(t)$ , divided by the temperature change at net zero,  $\Delta T(t_{ZE})$ . This estimate of the temperature change at net zero is performed using a 20 year averaging window around the time of net zero to reduce the effect of interannual variability. The averaging is performed on the 1% branch experiment that always includes carbon emissions with an approximately linear rise in temperature, rather than combining a forced response up to net zero and an unforced response after net zero; this choice follows MacDougall et al. (2020) to avoid a possible bias in the estimate of the temperature at net zero,  $\Delta T(t_{ZE})$ . This averaging approach is applied for all the variables evaluated at net zero in our normalised framework.

The temperature response after emissions defining the ZEC involves a variety of competing drivers (Fig. 2) involving changes in carbon inventories, radiative forcing, radiative response and planetary heat uptake. These changes are next described and our framework applied to quantify the relative importance of these competing drivers.

### 2.4.1 Changes in carbon inventories

The carbon emissions lead to an increase in the atmospheric, ocean and carbon inventories: a temporary increase in the atmospheric carbon inventory (with a model mean and inter-model standard deviation) of 488±33 PgC at years 55-75 and the remainder taken up by the land and ocean inventories, 253±53 PgC and 207±26 PgC respectively (Fig. 2a-c). Post emissions, the cumulative carbon emission of 1006±31 PgC is more equally partitioned between the atmosphere, land and ocean, each holding 34%, 35% and 31% respectively of the emitted carbon at years 140-160 (typically years 70 to 90 after net zero).

#### 2.4.2 Changes in radiative response and planetary heat uptake

The changes in atmospheric carbon dioxide drive the changes in radiative forcing, reaching a maximum radiative forcing of  $3.1\pm0.2~\rm W~m^{-2}$  either at or within a year of the time of net zero (Fig. 2d). Most of the radiative forcing is returned to space with the radiative response reaching  $-2.0\pm0.5~\rm W~m^{-2}$  and a smaller planetary heat uptake reaching  $1.1\pm0.6~\rm W~m^{-2}$  (Fig. 2e,f). Post emissions, the radiative forcing reduces to  $2.4\pm0.3~\rm W~m^{-2}$  at years 140-160 with the radiative response only slightly decreasing in magnitude to  $-1.9\pm0.4~\rm W~m^{-2}$  and the planetary heat uptake reducing further to  $0.5\pm0.5~\rm W~m^{-2}$ .

Hence, the temperature response up to and after net zero involves changes in atmospheric carbon due to the land and ocean carbon uptake, and the resulting radiative forcing is either returned to space or used to warm the planet. The goal now is to

Figure 2. Climate response during emissions and post emissions versus time (year) since the pre industrial for the 9 Earth system models: changes in (a) atmospheric carbon inventory,  $\Delta I_A$  (PgC); (b) land carbon inventory,  $\Delta I_L$  (PgC); (c) ocean carbon inventory,  $\Delta I_O$  (PgC); (d) radiative forcing supplying heat to the climate system, F (W m<sup>-2</sup>); (e) radiative response representing a heat loss to space,  $-\Delta R$  (W m<sup>-2</sup>); and (f) planetary heat uptake,  $\Delta N$  (W m<sup>-2</sup>), positive representing a gain in heat. The plot includes smoothing of planetary heat uptake with a 10 year running mean.

draw upon the identity for the geometric ZEC in order to compare the effect of changes in the carbon sinks, radiative response and planetary heat uptake.

### 2.5 Controls of the ZEC and geometric ZEC

The ZEC measures the temperature change after net zero. The timing of net zero varies from years 61 to 71 in the set of models and, in our subsequent analysis, we choose to align their time series so that the timing of net zero coincides. The ZEC, defined by  $\Delta T(t) - \Delta T(t_{ZE})$ , reaches -0.04±0.14 K for year 25, and -0.11±0.19 K and -0.12±0.24 K for years 50 and 90 after net zero (Fig. 3a; Tables 1 and A1 for individual models) (MacDougall et al., 2020).

Alternatively, the geometric ZEC, given by the ratio of the temperature change relative to the pre industrial,  $\Delta T(t)$ , and the change for net zero,  $\Delta T(t_{ZE})$ , varies from  $0.97\pm0.09$  for year 25 to  $0.93\pm0.11$  and  $0.92\pm0.14$  respectively for years 50 and 90 after net zero (Table 1; Fig. 3b). The model-mean changes in the geometric ZEC are relatively small, accounting for a temperature anomaly decrease of only 8% after net zero. However, the individual model responses are much larger, reaching 20% changes after net zero; as previously highlighted by MacDougall et al. (2020).

The ZEC response is made up of competing responses that are quantified in the normalised framework (19): (i) the normalised thermal contribution,  $\Delta T(t)/\Delta F(t)$ , is large and positive, reaching  $1.22\pm0.11$  and  $1.33\pm0.15$  after 50 and 90 years respectively (Table 1; Fig. 3c); and (ii) the normalised radiative contribution,  $\Delta F(t)/\Delta I_A(t)$ , is relatively small, only reaching  $1.09\pm0.02$  and  $1.11\pm0.02$  after 50 and 90 years respectively (Fig. 3d); and (iii) the normalised carbon contribution,  $\Delta I_A(t)$ , is large and negative, reaching  $0.69\pm0.05$  and  $0.62\pm0.06$  after 50 and 90 years respectively (Table 1; Fig. 3e). Hence, the geometric ZEC is primarily determined by a competition between the normalised thermal and carbon contributions.

For individual models, there are some large variations, with the normalised thermal contribution exceeding a 30% increase for CESM2, CNRM-ESM2 and UKESM1, and the normalised carbon contribution reaching a 30% decrease for CanESM5, CESM2, CNRM-ESM2, GFDL-ESM2, and NorESM2 (Fig. 4, red and green lines; Table A1).

The resulting geometric ZEC response involves an interplay of these normalised thermal and carbon contributions. For example, the positive ZEC response for UKESM1 is due to a strong thermal contribution and only a moderate opposing carbon contribution, while the positive ZEC response for CNRM-ESM2 involves a very strong thermal contribution and an opposing strong carbon contribution. Meanwhile the negative ZEC response for NorESM2 is due to a relatively modest thermal contributions and relatively strong opposing carbon contributions.

The inter-model spread of the geometric ZEC, measured by the coefficient of variation, reaches 0.12 after 50 years and is made up of contributions of 0.09 for the thermal contribution, 0.02 for the radiative contribution and 0.07 for the carbon contribution (Table 1). Hence, the thermal contribution is the most important contributor to the inter-model spread in the geometric ZEC, closely followed by the carbon contribution and the radiative contribution is least important.

These competing carbon and thermal contributions for the ZEC are next addressed in more detail.

# 2.6 Carbon contribution to the ZEC response

The carbon contribution to the geometric ZEC response involves a normalised decrease in the atmospheric carbon inventory,  $\Delta I_A(t)$ , which is achieved by an increase in both land and ocean carbon inventories.

Figure 3. Temporal evolution of the temperature response, the ZEC and its components after net zero when emissions cease: (a) the surface temperature change,  $\Delta T(t')$  in K, after net zero is reached (year); (b) the ZEC, surface temperature change,  $\Delta T(t') - \Delta T(t'_{ZE})$  in K, after net zero is reached (year); (c) the geometric ZEC,  $\Delta T(t')/\Delta T(t'_{ZE})$ , a value greater than 1 defines a positive ZEC and a value less than 1 defines a negative ZEC; (d) the thermal contribution from the normalised dependence of surface temperature on radiative forcing,  $\Delta T(t')/\Delta F(t')$ ; (e) the radiative contribution from the normalised dependence of radiative forcing on atmospheric carbon,  $\Delta F(t')/\Delta I_A(t')$ ; (f) the carbon contribution from the normalised atmospheric carbon,  $\Delta I_A(t')$ . The time series for each individual model is aligned so that the timing of net zero coincides. The normalisation is taken from the average value of the variable over a 20 year period centered on net zero based on the linear response of the 1pct continually-forced experiment. The plot includes smoothing of temperature with a 10 year running mean.

Figure 4. Temporal evolution of the geometric ZEC and its contributions for 9 different Earth system models: the normalised surface temperature change,  $\Delta T(t')$ , (black line); the thermal contribution from the normalised dependence of surface temperature on radiative forcing,  $\Delta T(t')/\Delta F(t')$  (red line); the radiative contribution from the normalised dependence of radiative forcing on atmospheric carbon,  $\Delta F(t')/\Delta I_A(t')$  (blue line); and carbon contribution from the normalised atmospheric carbon,  $\Delta I_A(t')$  (green line). The normalisation is based on the values of each variable at the time of net zero. The plot includes smoothing of temperature, the alignment of the time series and normalisation as in Fig. 3.

In order to compare these different carbon sinks, the carbon changes of each inventory are henceforth normalised by the same cumulative carbon emission at net zero, as given by the airborne, landborne and oceanborne fractions. Each of these fractions are evaluated at a particular time using a 20 year time window centered on that time. The airborne fraction,  $\Delta I_A(t)/I_{em}(t_{ZE})$ , is a maximum at net zero and then declines in time for all models (Fig. 5, black line) due to the increase in the landborne and oceanborne fractions (Fig. 5, green and blue lines). The airborne fraction is  $0.52\pm0.03$  at net zero and decreases to  $0.38\pm0.05$  and  $0.34\pm0.05$  at years 50 and 90 after net zero (Table 1). The landborne fraction,  $\Delta I_L(t)/I_{em}(t_{ZE})$ , increases

Figure 5. Temporal evolution of cumulative airborne fraction,  $\Delta I_A(t')$  (black), landborne fraction  $\Delta I_L(t')$  (green) and oceanborne fraction  $\Delta I_O(t')$  (blue) for time (year) relative to net zero for 9 different Earth system models.

from  $0.26\pm0.04$  at net zero reaching  $0.34\pm0.07$  and  $0.35\pm0.08$  for 50 and 90 years later respectively; and the oceanborne fraction,  $\Delta I_O(t)/I_{em}(t_{ZE})$  increases from  $0.22\pm0.03$  at net zero reaching  $0.28\pm0.04$  and  $0.31\pm0.04$  for 50 and 90 years later. Hence, initially after net zero, the carbon uptake by the terrestrial system dominates over that by the ocean for most models, but they become comparable to each other by 90 years.

The landborne fraction is much larger than the oceanborne fractions for CanESM5, CNRM-ESM2 and GFDL-ESM2, while the landborne and oceanborne fractions are comparable for UKESM1 and the landborne fraction is much smaller than the oceanborne fraction for ACCESS-ESM1.5 (Fig. 5; Table A1).

These different relative strengths of the land and ocean carbon sinks are likely due to structural differences in the land carbon model. The three models with higher landborne fractions may be overestimating the possible land carbon sink as they neglect

the role of nutrient limitations. The other models include land nitrogen cycling and limitation of carbon allocation, and this nutrient limitation reduces carbon-cycle feedbacks (Arora et al., 2019). In a similar manner, the nitrogen limitation on land acts to reduce the land carbon sink and so slightly increase the resulting ZEC.

The inter-model spread is much larger for the landborne fraction than the oceanborne fraction with coefficients of variation of 0.21 and 0.13 respectively after 50 years (Table 1). These inter-model differences in the landborne and oceanborne fractions are partly compensating, since both coefficients of variation are larger than that for the airborne fraction reaching 0.12.

#### 2.7 Thermal response

The thermal contribution to the ZEC may be understood in terms of the top of the atmosphere energy balance (8). The radiative forcing,  $\Delta F(t)$ , peaks close to the time of net zero and then declines for each model (Fig. 6, black line).

The radiative response,  $\Delta R(t)$ , is negative and so represents the part of the radiative forcing that is returned to space. The radiative response varies between models, most involve a peak in magnitude at the time of net zero and then a slight decline in magnitude (such as CESM2 and NorESM2-LM), while in some models (such as CNRM-ESM2 and UKESM1) the radiative response remains relatively constant in time (Fig. 6, red line). The planetary heat uptake,  $\Delta N(t)$ , represents the mismatch between the radiative forcing and radiative response. The planetary heat uptake is a maximum at the time of net zero and declines in time for all models (Fig. 6, blue line). For these thermal quantities there is significant interannual variability.

The thermal contribution to the geometric ZEC response, the normalised  $\Delta T(t)/\Delta F(t)$ , may be derived from the top of the atmosphere energy balance (9). This thermal contribution increases after net zero (Fig. 7, black line) and is made up itself by the product of contributions from the fraction of the radiative forcing escaping to space, the normalised  $\Delta R(t)/\Delta F(t)$ , and the inverse of the climate feedback parameter, the normalised  $\lambda(t)^{-1}$  (Fig. 7, blue and red lines respectively).

The normalised fraction of the radiative forcing escaping to space,  $\Delta R(t)/\Delta F(t)$ , increases after net zero and reaches  $1.25\pm0.11$  and  $1.30\pm0.16$  for 50 and 90 years later respectively (Table 1, Fig. 7, red line). The normalised inverse of the climate feedback parameter,  $\lambda(t)^{-1}$ , is close to 1, reaching  $0.98\pm0.06$  and  $1.02\pm0.05$  for 50 and 90 years later respectively. Thus, the dominant contribution to the increase in the thermal contribution to the geometric ZEC response is from an increase in the fraction of the radiative forcing escaping to space, which is equivalent to a decrease in the fraction of radiative forcing used to increase planetary heat.

For most individual models, the thermal contribution to the geometric ZEC response, the normalised  $\Delta T(t)/\Delta F(t)$ , is broadly the same as the fraction of radiative forcing escaping to space,  $\Delta R(t)/\Delta F(t)$  (Fig. 7, blue line; Table A1). However, there is an enhancement of the thermal contribution from the increase in the radiative forcing escaping to space by a time-varying amplification from the climate feedback parameter for UKESM1 and for the latter parts of the temporal record for GFDL-ESM2 and NorESM2-LM (Fig. 7, red line).

Intermodel differences in the thermal contribution are dominated by differences in the fraction of radiative forcing returned to space, rather than from differences in the inverse of the climate feedback parameter, since their coefficients of variation are 0.09 and 0.06 respectively after 50 years (Table 1).

Figure 6. Temporal evolution of radiative forcing,  $\Delta F(t')$  (black), planetary heat uptake,  $\Delta N(t')$  (blue), and radiative response,  $-\Delta R(t')$  (red) in W m<sup>-2</sup> for time (year) relative to net zero for 9 different Earth system models. The plot includes smoothing of temperature and planetary heat uptake with a 10 year running mean.

These diagnostics of the normalised ZEC framework are based on single model ensembles from ZECMIP. The diagnostic framework is next applied to a much larger set of model ensembles to span more fully parameter space and reveal how representative our diagnostics for ZECMIP are.

# 3 Analyses of a large ensemble of an efficient Earth system model

## 3.1 Experiments with the efficient Earth system model

The ZEC diagnostics are now repeated for a large ensemble of an efficient Earth system model (WASP) (Goodwin, 2016).

Figure 7. Temporal evolution of the thermal contribution defined by the normalised temperature dependence on radiative forcing,  $\Delta T(t')/\Delta F(t')$  (black), normalised fraction of radiative forcing escaping to space,  $\Delta R(t')/\Delta F(t')$  (blue), and normalised reciprocal of climate feedback parameter  $\lambda(t')^{-1}$  (red) for time (year) relative to net zero for 9 different Earth system models. The alignment of the time series and normalisation as in Fig. 3. The plot includes smoothing of temperature and planetary heat uptake with a 10 year running mean.

WASP includes air-sea exchange of CO<sub>2</sub> includes a full carbonate chemistry solver for the surface ocean (Follows et al., 2006). Sub-surface ocean boxes then exchange carbon with the surface ocean with each sub-surface box having an e-folding timescale prescribed over which the sub-surface box becomes chemically equilibrated with the surface ocean. The land carbon cycle in WASP is separated into a vegetation carbon pool and a soil carbon pool. The net primary production removes carbon from the atmosphere into the vegetation pool. Net primary production is dependent upon atmospheric CO<sub>2</sub> via a logarithmic relationship using a CO<sub>2</sub>-fertilisation coefficient, and net primary production is sensitive to global mean temperature via a net primary production-temperature coefficient. The carbon flux from the vegetation to soil carbon pools is via leaf litter, which is

only dependent upon the size of the vegetation pool. The soil carbon pool returns carbon to the atmosphere with an e-folding timescale, which is temperature dependent via a third coefficient. WASP includes time-varying climate feedbacks, to represent time-varying changes in the pattern effect (Goodwin, 2018; Goodwin et al., 2020)

In these experiments, 10 million prior simulations are integrated using historical forcing and following the SSP245 experiment from year 2014 with varied model parameters (Supplementary Table S1; (Goodwin, 2021; Goodwin and Cael, 2021)). In an initial prior ensemble the coefficients are varied independently. This prior ensemble is historically forced and compared to observational reconstructions. Only ensemble members with land and ocean carbon uptake that are in accord with historic observational reconstructions are retained in the final WASP ensemble (<1% of prior ensemble members). Of these simulations, 1138 posterior solutions are identified that satisfy observable quantities (Goodwin, 2018).

These 1138 posterior ensemble members are then integrated forward following two different experiments: (i) an annual 1% increase in atmospheric CO<sub>2</sub> with emissions ceasing at 1000 PgC (referred to as the 1pctCO<sub>2</sub> case as for ZECMIP) or (ii) a constant emission rate of 10 PgC yr<sup>-1</sup> for 100 years until there is 1000 PgC emitted (referred to as the flat10 case) (Sanderson et al., 2024). This comparison is included as flat10 is a scenario choice for CMIP7 and has the benefit of a more constant forcing regime.

In the 10 million historically forced prior simulations used to determine observational consistency the WASP model simulations include an imposed internal variability (Goodwin, 2018). This internal variability is turned off when the posterior simulations are then forced with idealised experiments.

#### 3.2 ZEC responses for the large ensemble model

320

The ZEC responses reveal a slight decrease in surface temperature after net zero for the median of the ensembles for both the 1pctCO2 and flat10 experiments (Fig. 8a,b, blue line). For the 1pctCO2 experiment, the median ZEC and the 5% to 95% ensemble range in brackets are -0.10 K (-0.47 K to 0.43 K) after 50 years, increasing in magnitude to -0.09 K (-0.56 K to 0.82 K) after 100 years (Fig. 8a; Supplementary Table S2). There is close agreement in these ZEC estimates with the ZECMIP model mean of -0.10 K at 50 years (Table 1) and the ZECMIP range is comparable to the 1 standard deviation range from WASP (Fig. 8, orange line and dark blue shading).

For the flat10 experiment, the median ZEC and the 5% to 95% ensemble range in brackets are slightly smaller: -0.06 K (-0.25 K to 0.41 K) after 50 years, increasing in magnitude to -0.07 K (-0.31 K to 0.73 K) after 100 years and further to -0.19 K (-0.46 K to 1.02 K) after 400 years (Fig. 8b, Supplementary Table S3). This slightly smaller magnitude response for flat10 is due to there being a stronger radiative forcing for the 1pctCO2 case as the forcing is more exponential.

There is some slight curvature in the ZEC responses between the 1pctCO2 and flat10 experiments with a greater range for the 1pctCO2 ensembles (Fig. 8c).

The changes in carbon inventories, radiative forcing, radiative feedback and planetary heat uptake (Figs. S1, S2; Supplementary Tables S2 and S3) vary in a broadly similar manner as for the ZECMIP diagnostics over 100 years (Figs. 1 and 2), although include a much greater ensemble spread and extend for much longer to 400 years.

**Figure 8.** Simulations for ZEC from a large model ensemble from WASP over time since net zero (year): (a) the 1pctCO2 experiment, including the median (line), 1-sigma range (dark shading) and 95% range (light shading) together with the maximum and minimum ZECMIP range; (b) the flat10 experiment; and (c) scatterplot comparing the ZEC responses for the 1pctCO2 and flat10 experiment for 20, 50 and 100 years for each model realisation.

#### 3.3 Geometric ZEC and normalised contributions

The geometric ZEC slightly decreases after net zero for the ensemble median to 0.92, 0.92 and 0.82 for years 50, 100 and 400 for the 1pctCO2 experiment, and 0.95, 0.94 and 0.83 for years 50, 100 and 400 after net zero for the flat10 experiment (Supplementary Tables S2 and S3). There is a wide inter-ensemble spread with the geometric ZEC varying from 0.71 to 1.50 at year 100 including a tail of ensembles with much higher geometric ZEC (Fig. 9a).

For each ensemble, the geometric ZEC value (Fig. 9a) equates to the product of the ensemble values for each of the ZEC contributions (Fig. 9b-d). The thermal contribution to the geometric ZEC, the normalised  $\Delta T/\Delta F$ , increases in time for all ensembles to a median of 1.24 (5% and 95% range of 0.97 to 1.75) after 100 years (Fig. 9b). The radiative contribution to the geometric ZEC, the normalised  $\Delta F/\Delta I_A$ , only slightly increases in time for all ensembles to 1.07 (1.04 to 1.09) after 100 years (Fig. 9c). The carbon contribution to the geometric ZEC given by the normalised change in atmospheric carbon,  $\Delta I_A$ , decreases for all ensembles to 0.71 (0.67 to 0.86) after 100 years (Fig 9d).

The thermal contribution to the geometric ZEC, the normalised  $\Delta T(t)/\Delta F(t)$  and its contributions, reveals similar changes as for median ensemble response as in ZECMIP (Fig. 10a, full lines): there is a strengthening in the normalised  $\Delta T(t)/\Delta F(t)$  in time, which is primarily due to the strengthening in the normalised fraction of radiative forcing returned to space,  $1 - \Delta N(t)/\Delta F(t)$  augmented by a strengthening in the normalised inverse climate feedback,  $\lambda(t)^{-1}$ .

Figure 9. Geometric ZEC and its normalised components following the 1pctCO2 (left) and flat10 (right) experiment from the WASP simulations: (a) the geometric ZEC from the surface temperature change divided by the value at net zero,  $\Delta T(t')/\Delta T(t_{ZE})$ , including the median (blue line), 1-sigma range (dark shading) and 95% range (light shading) and bounds from ZECMIP (black dashed line); (b) the thermal contribution from the normalised dependence of surface temperature on radiative forcing,  $\Delta T(t')/\Delta F(t')$ ; (c) the radiative contribution from the normalised dependence of radiative forcing on atmospheric carbon,  $\Delta F(t')/\Delta I_A(t')$ ; and (d) the carbon contribution from the normalised change in atmospheric carbon inventory,  $\Delta I_A(t')$ . In each case, the normalisation is by the value of the variable at the time of net zero.

This contribution is made up of the product of two terms, the normalised radiative response divided by the radiative forcing, the normalised  $\Delta R(t')/\Delta F(t')$ , and the normalised inverse of the climate feedback parameter, the normalised  $\lambda(t')^{-1}$ . There is a consistent increase in the fraction of the radiative forcing returned to space or equivalently a decrease in the fraction of radiative forcing used for planetary heat uptake with a relatively tight ensemble spread and the median increasing to 1.12 (5% and 95% spread of 1.00 to 1.19) (Fig. 10a, blue line and shading; Supplementary Tables S2 and S3). The normalised inverse of the climate feedback parameter only slightly increases in time for the median to 1.10 after 100 years (Fig. 10a, orange line and pale shading), but there is a wide, asymmetrical spread with the 5% to 95% range extending from 0.84 to 1.73, so including ensembles with much larger  $\lambda(t)^{-1}$  providing a greater amplification by climate feedbacks compared with the ZECMIP diagnostics.

The carbon contribution to the geometric ZEC, the normalised atmospheric carbon,  $\Delta I_A(t)$ , may be understood by the changes in airborne fraction,  $\Delta I_A(t)/I_{em}(t)$ , which progressively decreases in time (Fig. 10b, black line and grey shading; Supplementary Table S2 and S3). The dominant contribution to the changes in airborne fraction alters from being from the

Figure 10. Temporal evolution of thermal and carbon variables affecting the geometric ZEC for the 1pctCO2 (left) and flat10 (right) experiments from the WASP simulations extending to 300 years after the time of net zero: (a, b) thermal contribution to the geometric ZEC from the normalised temperature dependence on radiation,  $\Delta T(t')/\Delta F(t')$  (black line for median, grey shading for 95% range), normalised fraction of radiative forcing escaping to space,  $\Delta R(t')/\Delta F(t') = (1 - \Delta N(t')/\Delta F(t'))$  (blue line and shading) and normalised inverse climate feedback parameter  $\lambda(t')^{-1}$  (orange line and pale shading); and (c, d) the partitioning of cumulative carbon emissions into the airborne fraction (black line for median and grey shading for 95% range), oceanborne fraction (blue line and shading) and landborne fraction (green line and shading) for time (year) relative to net zero. The ZECMIP bounds are included as dashed lines.

landborne fraction close to the time of net zero,  $\Delta I_L(t)/I_{em}(t)$ , to the oceanborne fraction on timescales greater than 50 years after net zero,  $\Delta I_O(t)/I_{em}(t)$  (Fig. 10b, green and blue lines and shading respectively).

There is a much larger ensemble spread around the landborne and ocean borne responses than for the airborne fraction, which implies that the changes in land and ocean carbon sinks partly compensate for each other. This partial compensation in carbon sinks is consistent with the coefficient of variation being larger for the landborne and oceanborne fractions than the airborne fraction as diagnosed for ZECMIP (Table 1).

In summary, the ZEC responses and their normalised contributions are broadly similar in the diagnostics of the large ensemble WASP and the smaller set of 9 Earth system models in ZECMIP. The WASP assessment reveals partial compensation between changes in landborne and oceanborne fractions, and a larger spread in the effect of the climate feedback and the possibility of climate amplification of the ZEC.

#### 4 Discussion and Conclusions

The Zero Emissions Commitment (ZEC) measures whether there is an increase or decrease in global mean surface temperature after carbon emissions cease at the time of net zero. This temperature change after net zero represents a transient response to past carbon emissions. The climate response after emissions cease relative to the pre-industrial era may then be viewed in terms of (i) the global temperature rise associated with the amount of cumulative carbon emissions since the pre industrial, as measured by the Transient Climate Response to Cumulative CO<sub>2</sub> Emissions (TCRE), plus (ii) the subsequent transient temperature change due to prior carbon emissions, as measured by the Zero Emissions Commitment.

There are a wide range of climate processes that affect the ZEC and the transient climate response after net zero involving radiative forcing and the global cycling of carbon and heat. Gaining insight as to the relative importance of these different carbon and thermal processes in determining the ZEC is challenging due to their complexity and the effect of carbon and climate feedbacks (Palazzo Corner et al., 2023).

In order to gain mechanistic insight as to the controls of the ZEC, a normalised framework is developed that draws upon two fundamental balances: the top of the atmosphere energy budget (Gregory et al., 2004), representing how the planet warms in response to radiative forcing; and how carbon emissions are partitioned between the carbon inventories of the atmosphere, land and ocean (Jones et al., 2013). In our framework, firstly, a geometric ZEC is defined that measures the fractional zero emission commitment from the fraction of warming relative to the time of zero emissions. Secondly, the geometric ZEC is connected to the product of normalised thermal, radiative and carbon contributions, which depend upon respectively the dependence of surface temperature to radiative forcing, the dependence of radiative forcing on atmospheric carbon and the change in atmospheric carbon. Each of these contributions may then be interpreted in terms of underlying mechanisms: the thermal contribution connected to the top of the atmosphere energy balance; the radiative balance connected to the logarithmic dependence of radiative forcing on atmospheric CO<sub>2</sub>; and the carbon contribution connected to land and ocean carbon sinks.

Our normalised framework is applied to diagnostics of (i) 9 Earth system models following the ZECMIP protocols with a 1% annual increase in atmospheric CO<sub>2</sub> (1pctCO<sub>2</sub>) until a 1000 PgC cumulative carbon emission (Jones et al., 2019; MacDougall et al., 2020) and (ii) a large ensemble of an efficient Earth system model (Goodwin et al., 2020), which is applied to the same scenario and a scenario of a constant carbon emission (flat10) over 100 years until a 1000 PgC cumulative carbon emission (Sanderson et al., 2024). In both sets of diagnostics, the ZEC response is controlled by a competition between a cooling from a carbon contribution versus a warming from a thermal contribution. The carbon contribution involves the effects of land and ocean sinks in taking up carbon from the atmosphere. There is a strengthening in the magnitude of the carbon contribution in time with the land sink either increasing or saturating in time according to whether nitrogen cycling is included in the land sink, and the ocean sink increasing in time. The thermal contribution involves the dependence of surface temperature on radiative forcing. There is a strengthening in the thermal contribution with a larger fraction of the radiative forcing warming the surface and a smaller fraction being used for planetary heat uptake — this response is consistent with a declining efficiency in global ocean heat uptake and ventilation in time. The thermal contribution may also be augmented by the effect of climate feedbacks

that can amplify the surface warming, such as from a decrease in surface albedo or cloud albedo leading to an increase in solar absorption.

- There are detailed differences in the changes in the carbon and thermal contributions within the suite of Earth system models in ZECMIP:
  - (i) for the carbon contribution, some models (CESM2, CNRM-ESM2) have the land sink strengthening in time and always dominating over the ocean, while other models (ACCESS-ESM1.5, UKESM1) have the land sink saturating due to nitrogen limitation and the ocean sink eventually dominating the former response leads to a strengthening in the magnitude of the carbon contribution and acts to give a negative ZEC:
  - (ii) for the thermal contribution, some models (CESM2, NorESM2-LM) have planetary heat uptake and the radiative response declining in time after net zero, while other models (CNRM-ESM2, UKESM1) have the planetary heat uptake declining and the radiative response remaining nearly constant in time the latter response leads to a more marked strengthening in the thermal contribution and acts to give a positive ZEC;
- 415 (iii) the resulting ZEC response varies with these competing contributions, for example, a negative ZEC for NorESM2-LM is due to a large carbon uptake by the land and ocean, a positive ZEC for CNRM-ESM2 is due to a strong thermal contribution, while the positive ZEC for UKESM1 is due to a strong thermal contribution being reinforced by a more modest land carbon uptake.

Inter-model differences in these ZEC responses are primarily determined by differences in the radiative response, planetary heat uptake and land carbon uptake, and with more minor contributions from the ocean uptake of carbon and climate feedbacks. In comparison, inter-model differences in the TCRE are primarily controlled by thermal contributions (MacDougall et al., 2017) involving differences in climate feedbacks and ocean heat uptake, and then by carbon contributions involving the land uptake of carbon and a lesser extent the ocean uptake of carbon (Williams et al., 2020).

There are caveats and approximations in our ZECMIP analysis. The ZEC is a small signal:noise problem, and diagnosing this signal from Earth system models has inherent uncertainty (Borowiak et al., 2024). The ZECMIP diagnostics focus on a single model realisation and there are errors associated with how representative a single realisation is compared to a set of realisations by the same model. Our estimate of the radiative forcing from atmospheric CO<sub>2</sub> is based on a simple logarithmic closure and there are more accurate closures that may be applied. The estimate of the climate feedback is diagnosed from the radiative response from the energy balance at the top of the atmosphere divided by the changes in global surface temperature; this diagnostic is noisy on an interannual timescale. The Earth system models have inherent limitations in their representation of climate and carbon processes, especially involving uncertainties in cloud feedbacks, relatively coarse representation of ocean ventilation and a range of different land closures for carbon uptake. Finally, the ZECMIP analysis by design only includes the radiative forcing effects of atmospheric CO<sub>2</sub>, while in reality there are additional radiative forcing effects from other greenhouse gases and aerosols.

To partly address the above caveats, diagnostics of a large ensemble of an efficient Earth system model (WASP) are also examined and their ensemble median responses for the ZEC and its contributions are broadly similar to those of ZECMIP. The larger ensemble spread in WASP more clearly reveals a partial compensation between changes in the landborne and oceanborne

fractions, as well as the ocean carbon sink dominating on longer centennial timescales. The larger ensemble spread in WASP also reveals a wider range in the climate feedback parameter and how its temporal variation can lead to an amplification of surface warming and contributing to a positive ZEC. This difference suggests that the limited number of ZECMIP models may not be fully sampling the possible climate feedback responses compatible with historic warming. The efficient Earth system model does though show less variability in the heat uptake response, which may be due to a limitation that its ocean circulation is unchanging with time.

In summary, the geometric ZEC and normalised framework provide insight as to the reasons why there are different temperature responses from Earth system models after carbon emissions cease. Key processes are: whether the land carbon sink
saturates in time or continues to grow like the ocean sink after emissions cease, and whether the radiative response returning radiative forcing back to space declines in time or whether that radiative response remains nearly constant in time after
emissions cease, linked possibly to strengthening climate feedbacks. Gaining this process insight as to why the Earth system
models have a wide spread in their warming response after net zero is important for policy makers, since the combination of
the climate metrics, the ZEC and TCRE, affects estimates of how much carbon may be emitted before exceeding a warming
target.

Data availability. The data analysed for ZECMIP are openly available. The data input files are all based upon CMIP data that are available from the Earth System Grid Federation at https://esgf-node.llnl.gov/projects/esgf-llnl/. The data sets used for WASP will be achieved at Zenodo and a link provided to the WASP code.

Table 1. Statistics for the model-mean climate response and the inter-model spread for time relative to net zero (t', year): (a) Zero Emission Commitment (ZEC),  $\Delta T(t) - \Delta T(t_{ZE})$  and the geometric ZEC,  $ZEC/\Delta T(t_{ZE})$ , where  $\Delta$  is the change since the pre-industrial era and  $\Delta t_{ZE}$  is the temperature change at the time of net zero; (b) normalised contributions to the ZEC,  $\Delta T/\Delta F$  is the thermal contribution,  $\Delta F/\Delta I_A$  is the radiative contribution and  $\Delta I_A$  is the atmospheric carbon contribution; (b) normalised contributions to the thermal contribution,  $\Delta R/\Delta F$  is the fraction of radiative forcing escaping to space and  $\lambda^{-1}$  is the inverse of the climate feedback parameter; (d) Changes in the airborne, landborne and oceanborne fractions,  $\Delta I_A/I_{em}$ ,  $\Delta I_L/I_{em}$  and  $\Delta I_O/I_{em}$ . Model mean  $\overline{x}$ , inter-model standard deviation  $\sigma_x$  and coefficient of variation  $\sigma_x/\overline{x}$  are provided for the 9 CMIP6 models. For rows (b) to (d), the terms with a large normalised spread are underlined.

| time after net zero $t'$                                  | 25 y              | 50 y                                   | 90 y              |
|-----------------------------------------------------------|-------------------|----------------------------------------|-------------------|
| (a) ZEC                                                   |                   | $\Delta T(t) - \Delta T(t_{ZE})$ (K)   |                   |
| $\overline{x}\pm\sigma_x$                                 | $-0.04\pm0.14$    | $-0.11 \pm 0.19$                       | $-0.12 \pm 0.25$  |
| geometric ZEC                                             |                   | $\Delta T(t)/\Delta T(t_{ZE})$         |                   |
| $\overline{x}\pm\sigma_x$                                 | $0.97 \pm 0.09$   | $0.93 \pm 0.11$                        | $0.92 \!\pm 0.14$ |
| $\sigma_x/\overline{x}$                                   | 0.09              | 0.12                                   | 0.15              |
| (b) normalised contributions to the ZEC                   |                   |                                        |                   |
| thermal contribution                                      |                   | normalised $\Delta T(t)/\Delta F(t)$   |                   |
| $\overline{x}\pm\sigma_x$                                 | $1.16 \pm 0.09$   | $1.22 \pm 0.11$                        | $1.33 \pm 0.15$   |
| $\sigma_x/\overline{x}$                                   | 0.08              | <u>0.09</u>                            | <u>0.11</u>       |
| radiative contribution                                    |                   | normalised $\Delta F(t)/\Delta I_A(t)$ |                   |
| $\overline{x}\pm\sigma_x$                                 | $1.06 \pm 0.01$   | $1.09 \pm 0.02$                        | $1.11 \pm 0.02$   |
| $\sigma_x/\overline{x}$                                   | 0.01              | 0.01                                   | 0.02              |
| atmospheric carbon contribution                           |                   | normalised $\Delta I_A(t)$             |                   |
| $\overline{x}\pm\sigma_x$                                 | $0.79 \pm 0.04$   | $0.69 \pm 0.05$                        | $0.62 {\pm} 0.06$ |
| $\sigma_x/\overline{x}$                                   | 0.05              | 0.07                                   | 0.09              |
| (c) normalised contributions to $\Delta T(t)/\Delta F(t)$ |                   |                                        |                   |
| fraction of radiative forcing returned to space           |                   | normalised $\Delta R(t)/\Delta F(t)$   |                   |
| $\overline{x}\pm\sigma_x$                                 | $1.20 \pm 0.08$   | $1.25 \pm 0.11$                        | $1.30 \pm 0.16$   |
| $\sigma_x/\overline{x}$                                   | 0.07              | <u>0.09</u>                            | 0.12              |
| inverse climate feedback                                  |                   | normalised $\lambda(t)^{-1}$           |                   |
| $\overline{x}\pm\sigma_x$                                 | $0.96 \pm 0.04$   | $0.98 {\pm} 0.06$                      | $1.02 \pm 0.05$   |
| $\sigma_x/\overline{x}$                                   | 0.04              | 0.06                                   | 0.04              |
| (d) carbon changes                                        |                   |                                        |                   |
| time after net zero $t'$ 0 y                              | 25 y              | 50 y                                   | 90 y              |
| airborne fraction                                         |                   | $\Delta I_A(t)/I_{em}(t_{ZE})$         |                   |
| $\overline{x} \pm \sigma_x$ 0.52 $\pm$ 0.03               | $0.43 \pm 0.04$   | $0.38 {\pm} 0.05$                      | $0.34 {\pm} 0.05$ |
| $\sigma_x/\overline{x}$ 0.06                              | 0.10              | 0.12                                   | 0.14              |
| landborne fraction                                        |                   | $\Delta I_L(t)/I_{em}(t_{ZE})$         |                   |
| $\overline{x} \pm \sigma_x$ 0.26±0.04                     | $0.32 \pm 0.06$   | $0.34 \pm 0.07$                        | $0.35 {\pm} 0.08$ |
| $\sigma_x/\overline{x}$ 0.17                              | 0.20              | <u>0.21</u>                            | 0.21              |
| oceanborne fraction                                       |                   | $\Delta I_O(t)/I_{em}(t_{ZE})$         |                   |
| $\overline{x} \pm \sigma_x$ 0.22 $\pm$ 0.03               | $0.25 {\pm} 0.03$ | $0.28 \pm 0.04$                        | $0.31 \pm 0.04$   |
| $\sigma_x/\overline{x}$ 0.12                              | 0.13              | 0.13                                   | 0.14              |

### 455 5 Appendix

Table A1 displays the diagnostics for the individual 9 Earth system models making up ZECMIP at 50 years after net zero for the ZEC, the geometric ZEC and the thermal, radiative and carbon contributions.

Author contributions. RW developed the framework and wrote the main draft of the paper. RW diagnosed the ZECMIP models using model data provided by AM. CJ advised on the ZECMIP protocols and PC advised on the climate feedback diagnostics. PG performed the WASP integrations and diagnostics. All authors edited and commented on the manuscript.

Competing interests. There are no competing interests.

Acknowledgements. The authors acknowledge the World Climate Research Programme's Working Group on Coupled Modelling responsible for CMIP. RGW acknowledge support from the UK Natural Environmental Research Council, NE/N009789/1 and NE/W009501/1. PC was supported by UK Natural Environmental Research Council grants NE/V012045/1, NE/T006250/1 and EP/Y036123/1. CDJ was supported by the Joint UK BEIS/Defra Met Office Hadley Centre Climate Programme (GA01101) and the European Union's Horizon 2020 research and innovation programme under Grant Agreement No 101003536 (ESM2025 - Earth System Models for the Future).

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

**Table A1.** Climate model response post emissions at 50 years (40-59 year average) after net zero: (a) Zero Emission Commitment (ZEC),  $\Delta T(t) - \Delta T(t_{ZE})$ , and geometric ZEC,  $\Delta T(t)/\Delta T(t_{ZE})$ , where  $\Delta$  is the change since the pre-industrial era and  $\Delta t_{ZE}$  is the temperature change at net zero; (b) Normalised changes in the land and ocean carbon inventories,  $\Delta I_L$  and  $\Delta I_O$ ; (c) normalised contributions to the geometric ZEC,  $\Delta T/\Delta F$  is the thermal contribution,  $\Delta F/\Delta I_A$  is the radiative contribution and  $\Delta I_A$  is the atmospheric carbon contribution; (d) normalised contributions to the thermal contribution,  $\Delta R/F$  is the fraction of radiative forcing escaping to space and  $\lambda^{-1}$  is the inverse of the climate feedback parameter. Model mean  $\overline{x}$ , inter-model standard deviation  $\sigma_x$  and coefficient of variation  $\sigma_x/\overline{x}$  are included.

| ZEC         Model $\Delta T(t) - \Delta T(t_{ZE})$ (K)         ACCESS-ESM1.5 $0.00$ CanESM5 $-0.12$ CESM2 $-0.29$ CNRM-ESM2 $0.05$ GFDL-ESM2 $-0.27$ MIROC-ESL2 $-0.09$ MPI-ESM-2-LR $-0.23$ NOFESM2-LM $0.28$ $UKESM1$ $0.28$ $UKESM1$ $-0.11\pm0.19$ $\sigma_x/\bar{x}$ (c) norm. contrib. to geometric ZEC         thermal $0.28$ ACCESS-ESM1.5 $1.19$ CanESM5 $1.19$ CESM2 $1.17$ CNRM-ESM2 $1.36$ GFDL-ESM2 $1.11$ MIROC-ESL2 $1.11$ MROC-ESL2 $1.11$ NorESM2-LM $1.16$ NorESM2-LM $1.16$ NorESM2-LM $1.09$ | 7                    |                                   |                                |                                           |                                |
|------------------------------------------------------------------------------------------------------------------------------------------------------------------------------------------------------------------------------------------------------------------------------------------------------------------------------------------------------------------------------------------------------------------------------------------------------------------------------------------------------------------|----------------------|-----------------------------------|--------------------------------|-------------------------------------------|--------------------------------|
| SS-ESM1.5 M5 2 2-E-ESM2 -ESM2 C-ESL2 SM-2-LR M1 M1 M1 M1  M5 2 2 L-ESM2 C-ESL2 SS-ESM1.5 SS-ESM1.5 SS-ESM2 C-ESL2 SM-2-LR M3 M5 M5 M5 M5 M5 M5 M5 M5 M7 M1 M2 M2 M3 M3 M4 M3 M4 M4 M1                                                                                                                                                                                                                                                       | ر                    | geometric ZEC                     | airborne fraction              | landborne fraction                        | oceanborne fraction            |
| SS-ESM1.5 M5 2 1-ESM2 ESM2 ESM2 C-ESL2 SM-2-LR M1 M1 M1 M1 M2-LM M1  ESM2 C-ESL2 SS-ESM1.5 M5 M5 M5 M5 M5 M5 M5 M5 M5 M7 M1                                                                                                                                                                                                                                                                                                                                                  | $\Gamma(t_{ZE})$ (K) | $\Delta T(t)/\Delta T(t_{ZE})$    | $\Delta I_A(t)/I_{em}(t_{ZE})$ | $\Delta I_L(t)/I_{em}(t_{ZE})$            | $\Delta I_O(t)/I_{em}(t_{ZE})$ |
| M5 2 1-ESM2 E-ESM2 C-ESL2 SM-2-LR M1 M1 M1 M1 M1 M1 M2-LM M1 M2-LM M1 M5 M7 M1 M2-LM M1 M1 M1 M1                                                                                                                                                                                                                                                                                                                                                     | 00                   | 1.0                               | 0.46                           | 0.20                                      | 0.34                           |
| 2<br>-ESM2<br>-C-ESL2<br>SM-2-LR<br>M2-LM<br>M1<br>M1<br>M1<br>SS-ESM1.5<br>M5<br>-ESM2<br>-ESM2<br>-C-ESL2<br>SM-2-LR<br>M2-LM<br>M1                                                                                                                                                                                                                                                                                                                                                                            | 12                   | 0.94                              | 0.33                           | 0.43                                      | 0.24                           |
| ESM2ESM2ESM2 SM-2-LR M2-LM M1 M1 M1 M2-LM M5 SS-ESM1.5 M5 C-ESM2ESM2 SM-2-LR M2-LM M1 M1 M1                                                                                                                                                                                                                                                                                                                                                                                                                      | 59                   | 0.87                              | 0.38                           | 0.35                                      | 0.27                           |
| ESM2 C-ESL2 SM-2-LR M1 M1 M1 M1 M1 M2 M5 M5 C-ESM2 C-ESM2 C-ESM2 C-ESM2 M3 M3 M3-LM M1 M1                                                                                                                                                                                                                                                                                                                                                                                                                        | 5                    | 1.03                              | 0.39                           | 0.40                                      | 0.21                           |
| SM-2-LR M2-LM M1 M1 M1 M1 M2-LM M1  SS-ESM1.5 M5 C-ESM2 C-ESL2 SM-2-LR M2-LM M1                                                                                                                                                                                                                                                                                                                                                                                                                                  | 72                   | 0.79                              | 0.31                           | 0.40                                      | 0.29                           |
| SM-2-LR M2-LM M1 M1 SS-ESM1.5 M5 C-ESM2 C-ESM2 C-ESL2 SM-2-LR M2-LM M1                                                                                                                                                                                                                                                                                                                                                                                                                                           | 60                   | 0.93                              | 0.37                           | 0.35                                      | 0.28                           |
| M2-LM M1 M1 SS-ESM1.5 M5 L-ESM2 -ESM2 C-ESL2 SM-2-LR M2-LM M1                                                                                                                                                                                                                                                                                                                                                                                                                                                    | 23                   | 0.87                              | 0.36                           | 0.36                                      | 0.28                           |
| MI SS-ESM1.5 M5 L-ESM2 ESM2 C-ESL2 SM-2-LR M2-LM M1                                                                                                                                                                                                                                                                                                                                                                                                                                                              | 30                   | 0.78                              | 0.35                           | 0.34                                      | 0.31                           |
| SS-ESM1.5 M5 2 1-ESM2 ESM2 C-ESL2 SM-2-LR M2-LM                                                                                                                                                                                                                                                                                                                                                                                                                                                                  | 8;                   | 1.11                              | 0.43                           | 0.27                                      | 0.30                           |
| SS-ESM1.5 M5 2 1-ESM2 -ESM2 C-ESL2 SM-2-LR M2-LM                                                                                                                                                                                                                                                                                                                                                                                                                                                                 | -0.19                | $0.93\pm0.11$                     | $0.38\pm0.05$                  | $0.34\pm0.07$                             | $0.28\pm0.04$                  |
| 5-ESM1.5<br>15<br>ESM2<br>SM2<br>-ESL2<br>M-2-LR<br>12-LM                                                                                                                                                                                                                                                                                                                                                                                                                                                        |                      | 0.12                              | 0.12                           | 0.21                                      | 0.13                           |
| S-ESM1.5<br>IS<br>ESM2<br>SSM2<br>SSM2<br>SM2<br>SM2<br>SM2<br>SM2<br>SCL2<br>M-2-LR                                                                                                                                                                                                                                                                                                                                                                                                                             | o geometric ZEC      |                                   |                                | (d) thermal contribution to geometric ZEC |                                |
| 5-ESM1.5<br>15<br>ESM2<br>SM2<br>ESL2<br>M-2-LR<br>12-LM                                                                                                                                                                                                                                                                                                                                                                                                                                                         | nal                  | radiative                         | carbon                         | fraction of forcing returned to space     | inverse climate feedback       |
| S-ESM1.5<br>IS<br>ESM2<br>SM2<br>ESL2<br>M-2-LR<br>12-LM                                                                                                                                                                                                                                                                                                                                                                                                                                                         | $t)/\Delta F(t)$     | norm. $\Delta F(t)/\Delta I_A(t)$ | norm. $\Delta I_A(t)$          | norm. $\Delta R(t)/\Delta F(t)$           | norm. $\lambda(t)^{-1}$        |
| 15<br>ESM2<br>SSM2<br>ESL2<br>M-2-LR<br>12-LM                                                                                                                                                                                                                                                                                                                                                                                                                                                                    | 6                    | 1.06                              | 0.79                           | 1.17                                      | 1.02                           |
| ESM2<br>:SM2<br>:ESL2<br>M-2-LR<br>[2-LM                                                                                                                                                                                                                                                                                                                                                                                                                                                                         | 4                    | 1.10                              | 0.64                           | 1.33                                      | 1.00                           |
| ×                                                                                                                                                                                                                                                                                                                                                                                                                                                                                                                | 7                    | 1.10                              | 0.67                           | 1.28                                      | 0.91                           |
| w.                                                                                                                                                                                                                                                                                                                                                                                                                                                                                                               | 9                    | 1.09                              | 0.69                           | 1.47                                      | 0.92                           |
| ×                                                                                                                                                                                                                                                                                                                                                                                                                                                                                                                | 1                    | 1.10                              | 0.65                           | 1.23                                      | 0.90                           |
| ×                                                                                                                                                                                                                                                                                                                                                                                                                                                                                                                | 11                   | 1.08                              | 0.72                           | 1.14                                      | 1.06                           |
|                                                                                                                                                                                                                                                                                                                                                                                                                                                                                                                  | 9                    | 1.09                              | 0.69                           | 1.19                                      | 96.0                           |
|                                                                                                                                                                                                                                                                                                                                                                                                                                                                                                                  | 61                   | 1.11                              | 0.64                           | 1.14                                      | 96.0                           |
|                                                                                                                                                                                                                                                                                                                                                                                                                                                                                                                  | 8.                   | 1.07                              | 0.75                           | 1.33                                      | 1.05                           |
| $\overline{x} \pm \sigma_x \qquad 1.22 \pm 0.11$                                                                                                                                                                                                                                                                                                                                                                                                                                                                 | .0.11                | $1.09\pm0.02$                     | $0.69\pm0.05$                  | $1.25\pm0.11$                             | 0.98±0.06                      |
| $\sigma_x/\overline{x}$ 0.09                                                                                                                                                                                                                                                                                                                                                                                                                                                                                     | 61                   | 0.01                              | 0.07                           | 0.09                                      | 90:0                           |