# Peer review of "A normalised framework for the Zero Emissions Commitment"

_EGUsphere, 2025_

## Author Comment (AC1)

**Response to Biogeosciences**

We thank both referees for their constructive comments.

RC1

**Review of "A normalised framework for the Zero Emissions Commitment"**

This article explores the contributions of thermal, carbon cycle, and radiative components to the temperature response after net zero $CO_2$ emissions. The analysis uses nine ESMs from the ZECMIP A1 protocol and an efficient Earth system model (WASP), which includes 1138 posterior ensemble simulations. A normalised ZEC framework is applied, where the ZEC is the temperature change relative to pre-industrial compared to the change at net zero. The ZEC response is determined by a competition between cooling from declining atmospheric $CO_2$ and warming from strengthening thermal contributions, as ocean heat uptake declines and climate feedbacks amplify surface warming. Different models achieve positive or negative ZEC through varying strengths of thermal and carbon contributions; for example, a strong thermal contribution drives positive ZEC in CNRM-ESM2, while large land and ocean carbon uptake leads to negative ZEC in NorESM2-LM. Inter-model differences are mainly driven by variations in ocean heat uptake and land carbon uptake, and diagnostics from WASP suggest that current model ensembles may underestimate the range of possible climate feedback responses.

Overall, this article is of excellent quality and makes an important contribution by clarifying the physical mechanisms that govern model behaviour after net zero emissions. The revisions suggested below are minor and primarily aimed at improving clarity.

We thank the referee for the positive comments.

**General**

It would enhance clarity by explaining the interpretation of some of the values once calculated. For example, in lines 210–212 the values for the normalised ZEC is given. I know that the interpretation of the normalised ZEC is provided in line 92. However, because there are many different variables used in this article, an explanation also in lines 210-212 here would improve understandability. This goes for many of the values throughout.

It is important to note that we are not changing the existing definition of ZEC as an absolute temperature change relative to the point of zero emissions. But for our framework we introduce an additional metric of relative change. We are switching to defining the arithmetic (i.e. existing) ZEC as DT(t) -DT(t_ze), and a new geometric ZEC as DT(t)/DT(t_ze), giving the fractional zero emission commitment (measuring the fraction of warming relative to the time of zero emissions).

 For the geometric ZEC a value of 1 means that the arithmetic ZEC is 0, and a value of 0.97 means that there is a negative ZEC and that there is a 3% decrease in the temperature change compared with the temperature change at net zero.

**Line Specific**

Line 91: "A positive ZEC corresponds to $\Delta T (t)/\Delta T (tZE) > 0$ and a negative ZEC to $\Delta T (t)/\Delta T (tZE) < 0$."

Thank you. This is a slip as spotted by the referee, and these different regimes should be defined by 1 rather than 0.

Should this threshold be one rather than zero? A normalised ZEC less than one would indicate cooling relative to the net zero temperature, and greater than one would imply warming.

Correct

Additionally, moving this explanation to immediately follow the definition of the normalised ZEC would make more sense (line 85).

Agreed, and now call this the geometric ZEC measuring the fractional zero emission commitment (measuring the fraction of warming relative to the time of zero emissions).

Line 217: Values such as $\Delta T(t)/\Delta F(t)$ are given without units. Some values are given with units (such as ZEC), and some are given without units. This occurs throughout with several other values that I believe should also have units.

These variables have been normalised by their values at the time of net zero, so that these normalised variables do not have any units. The text does state that the normalised value is used.

Line 220: The explanation in this section would benefit from additional clarification. It appears that contributions are inferred from changes between years 50 and 90, but this is not explicitly stated. For example:

"The normalised carbon contribution, $\Delta IA(t)$, decreases from $0.70 \pm 0.06$ at year 50 to $0.63 \pm 0.06$ at year 90."

Clarifying the methodology used to calculate these contributions would make the reasoning more transparent.

Agreed that more explicit clarification is helpful. The atmospheric carbon inventory is defined at the time periods centred at the time of net zero and at years 50 and 90 years after net zero with all values evaluated relative to the pre industrial. For each time period, the average is taken over a time window centred on that time, so that the time period for year 50 is taken as an average of years 40 to 59 years. The normalised change in the atmospheric inventory is given by that value divided by the value at the time of net zero, $\Delta I_A(t)/\Delta I_A(t\_ze)$. These values are given in Table 1b.

To understand the changes in the carbon system, we find it useful to use the airborne fraction so normalise the atmospheric change in the carbon inventory by the cumulative carbon emission at the time of net zero. This choice enables a clearer comparison between the atmosphere, land and ocean contributions as shown in Table 1d and Figure 5. We will make our notation more explicit for the Table and figure.

Figures 2 and 3 The lines in these figures are difficult to distinguish, and the legend overlaps with the x-axis. Using more distinct colours or varied line styles would improve figure readability.

Agreed. We have provided Figures 2 and 3 in a new layout and modified the line colour for one of the models.

New versions of figures 2 and 3.

[Figure]

**Figure 2.** Climate response during emissions and post emissions versus time (year) since the pre industrial for the 9 Earth system models: changes in (a) atmospheric carbon inventory, $\Delta I_A$ (PgC); (b) land carbon inventory, $\Delta I_L$ (PgC); (c) ocean carbon inventory, $\Delta I_O$ (PgC); (d) radiative forcing supplying heat to the climate system, $F$ (W m$^{-2}$); (e) radiative response representing a heat loss to space, $-\Delta R$ (W m$^{-2}$); and (f) planetary heat uptake, $\Delta N$ (W m$^{-2}$), positive representing a gain in heat. The plot includes smoothing of planetary heat uptake with a 10 year running mean.

[Figure]

**Figure 3.** Temporal evolution of the temperature response, the ZEC and its components after net zero when emissions cease: (a) the surface temperature change, $\Delta T(t')$ in K, after net zero is reached (year); (b) the ZEC, surface temperature change, $\Delta T(t') - \Delta T(t'_{ZE})$ in K, after net zero is reached (year); (c) the geometric ZEC, $\Delta T(t')/\Delta T(t'_{ZE})$, a value greater than 1 defines a positive ZEC and a value less than 1 defines a negative ZEC; (d) the thermal contribution from the normalised dependence of surface temperature on radiative forcing, $\Delta T(t')/\Delta F(t')$; (e) the radiative contribution from the normalised dependence of radiative forcing on atmospheric carbon, $\Delta F(t')/\Delta I_A(t')$; (f) the carbon contribution from the normalised atmospheric carbon, $\Delta I_A(t')$. The time series for each individual model is aligned so that the timing of net zero coincides. The normalisation is taken from the average value of the variable over a 20 year period centered on net zero based on the linear response of the 1pct continually-forced experiment. The plot includes smoothing of temperature with a 10 year running mean.

---

## Author Response (AR1)

**Response to Biogeosciences**

We thank both referees for their constructive comments.

RC1

Review of "A normalised framework for the Zero Emissions Commitment"

This article explores the contributions of thermal, carbon cycle, and radiative components to the temperature response after net zero CO2 emissions. The analysis uses nine ESMs from the ZECMIP A1 protocol and an efficient Earth system model (WASP), which includes 1138 posterior ensemble simulations. A normalised ZEC framework is applied, where the ZEC is the temperature change relative to pre-industrial compared to the change at net zero. The ZEC response is determined by a competition between cooling from declining atmospheric CO2 and warming from strengthening thermal contributions, as ocean heat uptake declines and climate feedbacks amplify surface warming. Different models achieve positive or negative ZEC through varying strengths of thermal and carbon contributions; for example, a strong thermal contribution drives positive ZEC in CNRM-ESM2, while large land and ocean carbon uptake leads to negative ZEC in NorESM2-LM. Inter-model differences are mainly driven by variations in ocean heat uptake and land carbon uptake, and diagnostics from WASP suggest that current model ensembles may underestimate the range of possible climate feedback responses.

Overall, this article is of excellent quality and makes an important contribution by clarifying the physical mechanisms that govern model behaviour after net zero emissions. The revisions suggested below are minor and primarily aimed at improving clarity.

We thank the referee for the positive comments.

**General**

It would enhance clarity by explaining the interpretation of some of the values once calculated. For example, in lines 210–212 the values for the normalised ZEC is given. I know that the interpretation of the normalised ZEC is provided in line 92. However, because there are many different variables used in this article, an explanation also in lines 210-212 here would improve understandability. This goes for many of the values throughout.

It is important to note that we are not changing the existing definition of ZEC as an absolute temperature change relative to the point of zero emissions. But for our framework we introduce an additional metric of relative change. We are switching to defining the arithmetic (i.e. existing) ZEC as DT(t)-DT(t\_ze), and a new geometric ZEC as DT(t)/DT(t\_ze), giving the fractional zero emission commitment (measuring the fraction of warming relative to the time of zero emissions).

For the geometric ZEC a value of 1 means that the arithmetic ZEC is 0, and a value of 0.97 means that there is a negative ZEC and that there is a 3% decrease in the temperature change compared with the temperature change at net zero.

New text provided on L85.

Alternatively, a geometric measure of the ZEC is given by the ratio of the temperature change,  $\Delta T(t)$ , and the temperature change at the time of net zero,  $\Delta T(tZE)$ , and measures the fractional zero emission commitment,  $\Delta T(t)/\Delta T(tZE)$ . (5)

A positive ZEC corresponds to this geometric measure,  $\Delta T(t)/\Delta T(tZE) > 1$ , and a negative ZEC to  $\Delta T(t)/\Delta T(tZE) < 1$ .

**Line Specific**

Line 91: "A positive ZEC corresponds to  $\Delta T$  (t)/ $\Delta T$  (tZE) > 0 and a negative ZEC to  $\Delta T$  (t)/ $\Delta T$  (tZE) < 0."

Thank you. This is a slip as spotted by the referee, and these different regimes should be defined by 1 rather than 0.

L85

A positive ZEC corresponds to this geometric measure,  $\Delta T(t)/\Delta T(tZE) > 1$ , and a negative ZEC to  $\Delta T(t)/\Delta T(tZE)

Figure 2. Climate response during emissions and post emissions versus time (year) since the pre industrial for the 9 Earth system models: changes in (a) atmospheric carbon inventory,  $\Delta I_A$  (PgC); (b) land carbon inventory,  $\Delta I_L$  (PgC); (c) ocean carbon inventory,  $\Delta I_O$  (PgC); (d) radiative forcing supplying heat to the climate system, F (W m-2); (e) radiative response representing a heat loss to space,  $-\Delta R$  (W m-2); and (f) planetary heat uptake,  $\Delta N$  (W m-2), positive representing a gain in heat. The plot includes smoothing of planetary heat uptake with a 10 year running mean.

Figure 3. Temporal evolution of the temperature response, the ZEC and its components after net zero when emissions cease: (a) the surface temperature change,  $\Delta T(t')$  in K, after net zero is reached (year); (b) the ZEC, surface temperature change,  $\Delta T(t') - \Delta T(t'_{ZE})$  in K, after net zero is reached (year); (c) the geometric ZEC,  $\Delta T(t')/\Delta T(t'_{ZE})$ , a value greater than 1 defines a positive ZEC and a value less than 1 defines a negative ZEC; (d) the thermal contribution from the normalised dependence of surface temperature on radiative forcing,  $\Delta T(t')/\Delta T(t')$ ; (c) the radiative contribution from the normalised dependence of radiative forcing on atmospheric carbon,  $\Delta F(t')/\Delta I_A(t')$ ; (f) the carbon contribution from the normalised atmospheric carbon,  $\Delta I_A(t')$ . The time series for each individual model is aligned so that the timing of net zero coincides. The normalisation is taken from the average value of the variable over a 20 year period centered on net zero based on the linear response of the 1pct continually-forced experiment. The plot includes smoothing of temperature with a 10 year running mean.

Thank you for the above comments and queries, which hopefully have helped the clarity of the paper.

**RC2**

**General Comments**

This paper proposes framework for developing understanding of the drivers of the Zero Emissions Commitment (ZEC) by introducing a normalized ZEC. Normalized ZEC accounts for the warming that has already occurred at the time of zero emissions.

This additional climate metric appears potentially useful for assessing variability both across Earth system models and within a simpler Earth system model.

**We thank the referee for the positive comments.**

I think the paper would be improved by clarifying the intent of introducing Normalized ZEC. Are the authors arguing that normalized ZEC should be used instead of ZEC?

**There are two parts to our response:**

(i) We think that retaining the usual definition for the ZEC is useful, but that we are providing a framework to identify the drivers of the ZEC.

The important point is that our definition provides a simpler connection between the temperature change to the different drivers involving the top of the atmosphere energy budget, radiative forcing dependence and the carbon inventory changes.

If you wish to compare the relative importance of the different drivers for the temperature change to each other, then that comparison is clearest if each term is normalised.

**(i) Labelling of a normalised ZEC**

We agree that labelling our definition as a normalised ZEC leads to confusion as there are other choices for that normalisation (such as that suggested by yourself).

The existing ZEC (e.g. as per MacDougall et al) is defined as an arithmetic measure of the absolute change of global temperature in degrees compared to the time of zero emissions.

Instead our new definition (that we had called a normalised ZEC) is equivalent to a geometric measure of the ZEC, given by the fractional zero emission commitment (measuring the fraction of warming relative to the time of zero emissions).

**New text on L85**

Alternatively, a geometric measure of the ZEC is given by the ratio of the temperature change,  $\Delta T(t)$ , and the temperature change at the time of net zero,  $\Delta T(tZE)$ , and measures the fractional zero emission commitment,  $\Delta T(t)/\Delta T(tZE)$ . (5)

A positive ZEC corresponds to this geometric measure,  $\Delta T(t)/\Delta T(tZE) > 1$ , and a negative ZEC to  $\Delta T(t)/\Delta T(tZE) < 1$ .

Or that Normalized ZEC is useful for comparing drivers of ZEC?

Agreed that the normalised framework is useful for comparing the relative importance of the different drivers of the ZEC.

The authors don't actually do a comparison with the original ZEC formulation so it is a bit hard to decipher if the normalized ZEC introduced here provides new information.

The ZEC is given by DT(t)-DT(t ze).

The geometric ZEC is given by DT(t)/DT(t ze).

Our framework is explicitly designed to identify the drivers of the geometric ZEC, such that the product of the normalised thermal, radiative and carbon drivers are exactly the same as that as the geometric ZEC as in equation (20).

Figures 3 and 4 shows the relative importance of each of those drivers for the geometric ZEC, and that is detailed in Table 1 and Table A1.

So the manuscript is designed to provide the information needed to understand the drivers of the ZEC.

In more detail, the statistics included in Table 1 includes the coefficient of variation for the different terms. The comparison of those coefficients of variation then reveals the relative importance of the different drivers or the components of the system.

I can see that it might, but it would be helpful to have a more direct comparison, or other direct representation of the benefit of this new metric whatever the authors think that is.

The framework is designed to provide a quantitative measure of the different drivers of the ZEC. Without a quantitative measure, one is left making qualitative comparison of thermal and carbon effects when those variables are measured in different ways. Our framework provides a formal way of comparing the relative importance of each driver.

For example, for the ZECMIP diagnostics for the geometric ZEC, we find that

- (i) The intermodal spread in the geometric ZEC is primarily controlled by the intermodal spreads in the normalised thermal contribution and normalised atmospheric carbon concentration, rather than that of the normalised radiative forcing dependence on atmospheric CO2 (Table 1b);
- (ii) The intermodal spread of the normalised contribution to the warming dependence on radiative forcing is mainly determined in ZECMIP by the intermodal spread in the fraction of radiative forcing returned to space rather than that of the inverse climate feedback (Table 1c);
- (iii) The intermodal spread of the airborne fraction is mainly determined by the intermodal spread of the landborne fraction, rather than the oceanborne fraction (Table 1c).

These detailed inferences were not the same for the inter-model spread for the TCRE, where inter-model differences in the thermal contribution were most important and they were primarily associated with inter-model differences in the climate feedback parameter (Williams et al., 2020, ERL, doi:10.1088/1748-9326/ab97c9).

There is quite a range of behavior across models for the individual component contributions to normalized ZEC. Can the authors do more to discuss why?

Explaining why the different Earth system models respond in different ways is challenging, but we can provide more insight for the carbon cycle. What we have done is identify which drivers are responsible for those different responses and how they link to different responses in the top of the atmosphere energy balance or how the airborne fraction is controlled.

We find that the model responses separate into different classes. For the carbon response, the land carbon sink either continues to increase in time or saturates. These different responses appear to be linked to whether there is a nutrient cycle that can inhibit the ability of the land to take up unlimited carbon.

For the top of the atmosphere response, the radiative response (returning heat to space) either weakens in time or remains relatively constant. These different responses connect to differences in the time evolution of the strength of climate feedbacks. Here there is not a simple message in terms of the complexity of the representation of climate processes (especially cloud processes) in determining the radiative response.

The main conclusions seem to be that ZEC is a balance between ocean heat uptake rate and carbon uptake rate (intermodel spread driven mostly by land). Isn't this already the view reported in review papers like Pallazo Corner et al 2023?

The review paper by Pallazo Corner et al 2023 only provided that insight in a qualitative manner. This framework provides insight into the relative importance of different drivers in a quantitative manner.

For example for the inter-model spread in the climate responses, the framework reveals

- (i) The intermodal spread in the geometric ZEC is primarily controlled by the intermodal spreads in the normalised thermal contribution and normalised atmospheric carbon concentration, rather than that of the normalised radiative forcing dependence on atmospheric CO2 (Table 1b):
- (ii) The intermodal spread of the normalised contribution to the warming dependence on radiative forcing is mainly determined in ZECMIP by the intermodal spread in the fraction of radiative forcing returned to space rather than that of the inverse climate feedback (Table 1c);
- (iii) The intermodal spread of the airborne fraction is mainly determined by the intermodal spread of the landborne fraction, rather than the oceanborne fraction (Table 1c).

Can the authors provide further ideas about what might lead to the balance of thermal vs. carbon contributions to ZEC across models?

We think that crucial issues for differences in model responses for ZECMIP are

- (i) for the top of the atmosphere energy balance is whether the radiative response stays constant or weakens (links to changes in climate feedbacks) and
- (ii) for the carbon budget is whether the land carbon sink saturates in time and so the relative importance of the land and ocean carbon sinks. Whether the land carbon saturates in time is linked to whether there is nutrient limitation.

Text insert on L405

There are detailed differences in the changes in the carbon and thermal contributions within the suite of Earth system models

- (i) for the carbon contribution, some models (CESM2, CNRM-ESM2) have the land sink strengthening in time and always dominating over the ocean, while other models (ACCESS-ESM1.5, UKESM1) have the land sink saturating due to nitrogen limitation and the ocean sink eventually dominating the former response leads to a strengthening in the magnitude of the carbon contribution and acts to give a negative ZEC;
- (ii) for the thermal contribution, some models (CESM2, NorESM2-LM) have planetary heat uptake and the radiative response declining in time after net zero, while other models (CNRM-ESM2, UKESM1) have the planetary heat uptake declining and the radiative response remaining nearly constant in time the latter response leads to a more marked strengthening in the thermal contribution and acts to give a positive ZEC;
- (iii) the resulting ZEC response varies with these competing contributions, for example, a negative ZEC for NorESM2-LM is due to a large carbon uptake by the land and ocean, a positive ZEC for CNRM-ESM2 is due to a strong thermal contribution, while the positive ZEC for UKESM1 is due to a strong thermal contribution being reinforced by a more modest land carbon uptake.

Text change on L445

Key processes are: whether the land carbon sink

saturates in time or continues to grow like the ocean sink after emissions cease, and whether the radiative response returning radiative forcing back to space declines in time or whether that radiative response remains nearly constant in time after emissions cease, linked possibly to strengthening climate feedbacks.

**Specific Comments**

line 30 - given that this sentence is describing ZEC in general it would make sense to cite more recent analyses of ZEC from flat10MIP (Sanderson et al. 2024)

We add a sentence after line 30 about the ZEC response to flat10MIP.

L35 This ZEC response may also be modified by the emission scenario (Sanderson et al., 2024).

line 54 - I had trouble remembering that I represented Carbon for the entire paper. Is there a reason the symbol is I and not C?

*I* is used for the carbon inventory throughout the whole paper.

C is often used for concentration.

Equation 4 - I think it would be helpful to write this as ZEC = DeltaT(t)-DeltaT(tze) to make it clear that this is the typical definition of ZEC

Agreed, happy to be more explicit.

L82

The ZEC measures the temperature change relative to the pre industrial,  $\Delta T(t)$ , minus the temperature change at the time of net zero, tZE,  $\Delta T(tZE)$ , and is defined by

 $ZEC \equiv \Delta T(t) - \Delta T(tZE)$ . (4)

line 80 - I wondered at first why you used DetltT(t)/DeltaT(tze) rather than ZEC/DeltaT(tze) since that is how I would think of a normalization.

**Normalised warming relative to net zero**

We switch to calling this ratio a geometric measure of the ZEC. Our choice is not done in an arbitrary manner, but instead chosen to reveal the drivers via equation (6), and then enable these variables to be connected to the top of the atmosphere energy budget and a carbon inventory budget.

One could normalise is a range of ways, but our choice enables a cleaner connection to the drivers.

I can guess that the chosen definition in Eq5 looks cleaner, and since the two definitions are simply related it makes more sense to use the simpler form. I think it would be helpful to point this out to readers to make it clear. After being explicit about ZEC in Eq 4 I would suggest either writing out or describing briefly that ZEC/DeltaT(tze) = "normalized ZEC"-1

Decided not to call this a normalised ZEC, but instead a geometric measure of the ZEC. The geometric ZEC measures the fractional zero emission commitment (measuring the fraction of warming relative to the time of zero emissions).

L85 Alternatively, a geometric measure of the ZEC is given by the ratio of the temperature change,  $\Delta T$  (t), and the temperature change at the time of net zero,  $\Delta T$  (tZE),

line 192 - it would be useful to also add the % change for land vs. ocean sink

Agreed, add percentage changes as well as the absolute values.

L195 the atmosphere, land and ocean, each holding 34%, 35% and 31% respectively of the emitted carbon at years 140-160 (typically years 70 to 90 after net zero).

line 194 - It would be helpful for intuition if the years could be translated into emissions, even if models have a range at this time.

We think that the referee means refer to the time after emissions cease. Agree that choice of time after net zero is preferable.

L196 (typically years 70 to 90 after net zero).

line 198 - how close to net zero is the maximum radiative forcing?

The time of net zero is defined by the branch point in the model integrations (Line 170). The maximum atmospheric CO2 and radiative forcing often coincides with that branch point, but sometimes can differ by a year. This difference is probably due to interannual variability.

L198 a maximum radiative forcing of 3.1±0.2 W m−2 either at or within a year of the time of net zero

line 216 - ZEC is also made up of competing thermal and carbon responses. I'd suggest rephrasing to say ZEC is made of competing responses... which are easier to cleanly quantify in the normalized framework.

Agreed. Happy to modify text.

L217 The ZEC response is made up of competing responses that are quantified in the normalised framework (19):

line 223 - "For individual models, there are some large variations". This sentence then goes on to show that more than half of models fall into this category. Can the authors say anything more useful about what might drive these variations?

The subsequent analysis of the carbon contribution in section 2.6 and the thermal response in section 2.7 is designed to provide more information as to those different responses, including figure 5 revealing the model differences in airborne fraction and figure 6 model differences for the top of the atmosphere energy budget.

For the airborne fraction there are intermodal differences due to the presence of nutrient cycling limiting the land cycle in some models and not in others.

**Added**

L235 These competing carbon and thermal contributions for the ZEC are next addressed in more detail

L248. The landborne fraction is much larger than the oceanborne fractions for CanESM5, CNRM-ESM2 and GFDL-ESM2, while the landborne and oceanborne fractions are comparable for UKESM1 and the landborne fraction is much smaller than the oceanborne fraction for ACCESS-ESM1.5 (Fig. 5; Table A1).

These different relative strengths of the land and ocean carbon sinks are likely due to structural differences in the land carbon model. The three models with higher landborne fractions may be overestimating the possible land carbon sink as they neglect the role of nutrient limitations. The other models include land nitrogen cycling and limitation of carbon allocation, and this nutrient limitation reduces carbon-cycle feedbacks (Arora et al., 2019). In a similar manner, the nitrogen limitation on land acts to reduce the land carbon sink and so slightly increase the resulting ZEC.

L263 The radiative response varies between models, most involve a peak in magnitude at the time of net zero and then a slight decline in magnitude (such as CESM2 and NorESM2-LM), while in some models (such as CNRM-ESM2 and UKESM1) the radiative response remains relatively constant in time (Fig. 6, red line).

line 285 - The authors need to provide more information about how carbon, and in particular land carbon, is represented in WASP. Given that this carbon sink rate is a key component readers need a brief description of how it is represented.

The version of WASP used has carbon exchange between the atmosphere and surface ocean employing a numerical carbonate chemistry solver (Follows et al., 2006 <a href="https://doi.org/10.1016/j.ocemod.2005.05.004">https://doi.org/10.1016/j.ocemod.2005.05.004</a>). Sub-surface ocean boxes then exchange carbon with the surface ocean with each sub-surface box having an e-folding timescale prescribed over which the sub-surface box becomes chemically equilibrated with the surface ocean.

The land carbon cycle in WASP contains a vegetation carbon pool and a soil carbon pool. The Net Primary Production (NPP) removes carbon from the atmosphere into the vegetation pool. NPP is dependent upon atmospheric CO2 via a logarithmic relationship using a CO2-fertilisation coefficient, and NPP is sensitive to global mean temperature via an NPP-T coefficient. The vegetation carbon to soil carbon pool flux is via leaf litter, which is only dependent upon the size of the vegetation pool. The soil carbon pool returns carbon to the atmosphere with an e-folding timescale, which is t

L293 WASP includes air-sea exchange of CO2 includes a full carbonate chemistry solver for the surface ocean (Follows et al., 2006). Sub-surface ocean boxes then exchange carbon with the surface ocean with each sub-surface box having an e-folding timescale prescribed over which the sub-surface box becomes chemically equilibrated with the surface ocean. The land carbon cycle in WASP is separated into a vegetation carbon pool and a soil carbon pool. The net primary production removes carbon from the atmosphere into the vegetation pool. Net primary production is dependent upon atmospheric CO2 via a logarithmic relationship using a CO2-fertilisation coefficient, and net primary production is sensitive to global mean temperature via a netprimary production-temperature coefficient. The carbon flux from the vegetation to soil carbon pools is via leaf litter, which is temperature dependent via a third coefficient.

line 289 - which parameters are being perturbed to create this ensemble? What aspects of carbon cycle parameters are being perturbed?

The three land carbon coefficients (CO2 fertilisation, NPP-T and soil carbon residence timescale-T) and the ocean deep box timescales are varied between WASP ensemble members, leading to significant differences in the carbon cycle responses. In an initial prior ensemble the coefficients are varied independently. This prior ensemble is historically forced and compared to observational reconstructions. Only ensemble members that see historic land and ocean carbon uptake agree with historic observational reconstructions are used in the final WASP ensemble (<1% of prior ensemble members).

**In the main text**

L303 In these experiments, 10 million prior simulations are integrated using historical forcing and following the SSP245 experiment from year 2014 with varied model parameters (Supplementary Table S1; (Goodwin, 2021; Goodwin and Cael, 2021)). In an initial prior ensemble the coefficients are varied independently. This prior ensemble is historically forced and compared to observational reconstructions. Only ensemble members with land and ocean carbon uptake that are in accord with historic observational reconstructions are retained in the final WASP ensemble (<1% of prior ensemble members). Of these simulations, 1138 posterior solutions are identified that satisfy observable quantities (Goodwin, 2018)

**Added in the supplementary**

WASP includes a large set of ensembles to span parameter space, which are constructed by varying key variables within prescribed bounds (Supplementary Table S1). WASP ensemble members include perturbations in the three land carbon coefficients for the CO2 fertilisation, net

primary production-temperature relationship and soil carbon residence timescale (Goodwin, 2016) and the ocean deep box timescales (Supplementary Table S1), leading to significant differences in the carbon cycle responses. Other model parameters varied between the prior ensemble members include climate feedbacks, separated into Planck, fast and multi-decadal feedbacks (Goodwin, 2021), and parameters relating to how changes in atmospheric constituents induce radiative forcing (Goodwin and Cael, 2021). The observational consistency tests used to extract the final posterior ensemble are as described in Goodwin (2021), with the land carbon consistency tests as in Goodwin (2018).

**Together with a Supplementary Table**

Table S1. Parameters varied in the WASP prior ensemble and their distributions. The distributions are: U - uniform distribution with minimum and maximum range specified; N - normal distribution with mean and standard deviation stated; LN - lognormal distribution with mean and variance of the underlying normal distribution stated; SN - skew normal distribution with mean, standard deviation and skew parameter stated. The land carbon cycle parameters and distributions ( $\partial NPP/\partial T$ ,  $\gamma_k$  and  $\partial \tau_{soil}/\partial T$ ) are as described in Goodwin (2016). The climate feedback parameters and their distributions ( $\lambda_P$ ,  $\lambda_f$  and  $\lambda_{md}$ ) are as described in Goodwin (2021). All other parameters are as described in Goodwin and Cael (2021).

| Parameter                                                                         | Symbol                            | Units                             | Range                                                                        |
|-----------------------------------------------------------------------------------|-----------------------------------|-----------------------------------|------------------------------------------------------------------------------|
| Sensitivity of net primary productivity (NPP) to temperature                      | $\partial NPP/\partial T$         | $\rm PgC~\rm yr^{-1}K^{-1}$       | U(-5 to+1)                                                                   |
| CO 2 fertilisation coefficient                                         | $\gamma_k$                        |                                   | U(0 to 1)                                                                    |
| Sensitivity of soil carbon residence timescale to temperature                     | $\partial \tau_{soil}/\partial T$ | $yr K^{-1}$                       | U(-1.36 to 0.45)                                                             |
| Planck climate feedback                                                           | $\lambda_P$                       | ${\rm W}{\rm m}^{-2}{\rm K}^{-1}$ | $LN \left( \ln 3.3, \ln(1 + 0.1^2/3.3^2) \right)$                            |
| Fast climate feedbacks                                                            | $\lambda_f$                       | ${\rm W}{\rm m}^{-2}{\rm K}^{-1}$ | $LN(\ln \lambda_P, \ln 2)) - \lambda_P$                                      |
| Multidecadal climate feedbacks                                                    | $\lambda_{md}$                    | ${\rm W}{\rm m}^{-2}{\rm K}^{-1}$ | $LN\left(\ln(\lambda_P + \lambda_f), \ln 2\right) - (\lambda_P + \lambda_f)$ |
| Ratio of SST change to global mean surface temperature at equilibrium             |                                   |                                   | U(0.2  to  1.5)                                                              |
| Ratio of deep ocean $T$ to SST change at equilibrium                              |                                   |                                   | U(0.1  to  1.0)                                                              |
| Timescale for fast climate feedbacks                                              | $\tau_{fast}$                     | days                              | N(8.9, 0.4)                                                                  |
| Timescale for multidecadal climate feedbacks (pattern effect)                     | $\tau_{md}$                       | years                             | U(20 to 45)                                                                  |
| Timescale for surface ocean mixed layer to equilibrate with atmospheric chemistry | $\tau_{mixed}$                    | years                             | U(0.5  to  1.0)                                                              |
| Timescale for upper ocean to equilibrate with surface ocean mixed layer           | $\tau_{upper}$                    | years                             | U(20  to  80)                                                                |
| Timescale for intermediate ocean to equilibrate with surface ocean mixed layer    | $\tau_{inter}$                    | years                             | U(150  to  1000)                                                             |
| Timescale for deep ocean to equilibrate with surface ocean mixed layer            | $\tau_{deep}$                     | years                             | U(500 to 1500)                                                               |
| Timescale for bottom ocean to equilibrate with surface ocean mixed layer          | $\tau_{bottom}$                   | years                             | U(1000 to 2500)                                                              |
| Radiative forcing coefficient for CO 2                                 | $a_{CO2}$                         | $\mathrm{W}\mathrm{m}^{-2}$       | N(5.35, 0.27)                                                                |
| Dimensionless uncertainty in CH 4 radiative forcing                    |                                   |                                   | N(1.0, 0.07)                                                                 |
| Dimensionless uncertainty in N2O radiative forcing                                |                                   |                                   | N(1.0, 0.05)                                                                 |
| Dimensionless uncertainty in halocarbon radiative forcing                         |                                   |                                   | N(1.0, 0.05)                                                                 |
| Direct radiative forcing from SO x aerosols in 2010                    | Yaero:SOx                         | $\mathrm{W}\mathrm{m}^{-2}$       | N(-0.31, 0.11)                                                               |
| Direct radiative forcing from black carbon aerosols in 2010                       | $\gamma_{aero:BC}$                | $\mathrm{W}\mathrm{m}^{-2}$       | N(0.18, 0.07)                                                                |
| Direct radiative forcing from organic carbon aerosols in 2010                     | $\gamma_{aero:OC}$                | $\mathrm{W}\mathrm{m}^{-2}$       | N(-0.03, 0.01)                                                               |
| Direct radiative forcing from NMVOC aerosols in 2010                              | $\gamma_{aero:NMVOC}$             | $\mathrm{W}\mathrm{m}^{-2}$       | N(-0.06, 0.09)                                                               |
| Direct radiative forcing from NO x aerosols in 2010                    | $\gamma_{aero:NOX}$               | $\mathrm{W}\mathrm{m}^{-2}$       | N(-0.08, 0.04)                                                               |
| Direct radiative forcing from NH 3 aerosols in 2010                    | Yaero:NH3                         | $\mathrm{W}\mathrm{m}^{-2}$       | N(-0.08, 0.04)                                                               |
| Indirect radiative forcing from aerosols in 2010                                  | $R_{aci:2010}$                    | $\mathrm{W}\mathrm{m}^{-2}$       | SN(-0.55, 0.37,-2.0)                                                         |
| Radiative forcing coefficient for volcanic aerosols                               |                                   | $\mathrm{W}\mathrm{m}^{-2}$       | N(-10, 0.5)                                                                  |

figure 9 - panel d, should the y-axis label be \Delta I\_A and not \_a? corrected

We can only do that for the 1% CO2 experiments and not the flat10 experiments, as shown in Figure 8a. To address this concern, we have added extra figure panels to figures 9 and 10 to include the WASP experiments for the 1% CO2 experiments and include the intermodal spread from the full ESMs.

Figure 9. Geometric ZEC and its normalised components following the 1pctCO2 (left) and flat10 (right) experiment from the WASP simulations: (a) the geometric ZEC from the surface temperature change divided by the value at net zero,  $\Delta T(t')/\Delta T(t_{ZE})$ , including the median (blue line), 1-sigma range (dark shading) and 95% range (light shading) and bounds from ZECMIP (black dashed line); (b) the thermal contribution from the normalised dependence of surface temperature on radiative forcing,  $\Delta T(t')/\Delta F(t')$ ; (c) the radiative contribution from the normalised dependence of radiative forcing on atmospheric carbon,  $\Delta F(t')/\Delta I_A(t')$ ; and (d) the carbon contribution from the normalised change in atmospheric carbon inventory,  $\Delta I_A(t')$ . In each case, the normalisation is by the value of the variable at the time of net zero.

figure 9 - why such a low spread in carbon? What is being perturbed about the carbon cycle in the ensemble?

The WASP carbon cycle sees a relatively wide distribution, but the normalised carbon cycle sees a relatively narrow distribution (Figure 9d). This is likely because the carbon cycle coefficients (e.g. CO2 fertilisation, NPP-T etc) are varied between ensemble members but within each ensemble member are held constant in time. So, when normalisation occurs to the point of zero emission, if a WASP ensemble member has had high anthropogenic carbon uptake up to that point then it will likely continue with high carbon uptake into the future.

figure 10 - it would be particularly helpful to put the intermodel spread (or the individual models? onto panel b so readers can compare the large spread in ESMs to the spread in WASP.

Agreed, we have added extra figure panels to figures 9 and 10 to include the WASP experiments for the 1% CO2 experiments, which include the intermodal spread from the full ESMs.

Figure 10. Temporal evolution of thermal and carbon variables affecting the geometric ZEC for the 1pctCO2 (left) and flat10 (right) experiments from the WASP simulations extending to 300 years after the time of net zero: (a, b) thermal contribution to the geometric ZEC from the normalised temperature dependence on radiation,  $\Delta T(t')/\Delta F(t')$  (black line for median, grey shading for 95% range), normalised fraction of radiative forcing escaping to space,  $\Delta R(t')/\Delta F(t') = (1 - \Delta N(t')/\Delta F(t'))$  (blue line and shading) and normalised inverse climate feedback parameter  $\lambda(t')^{-1}$  (orange line and pale shading); and (c, d) the partitioning of cumulative carbon emissions into the airborne fraction (black line for median and grey shading for 95% range), oceanborne fraction (blue line and shading) and landborne fraction (green line and shading) for time (year) relative to net zero. The ZECMIP bounds are included as dashed lines.

line 347 - "coefficient of variation being larger for the landborne and oceanborne fractions than the airborne fraction" - I'd like the authors to discuss what about the model structure of WASP could cause this?

There are two reasons for this "coefficient of variation being larger for the landborne and oceanborne fractions than the airborne fraction", one to do with the WASP ensemble and another to do with the system itself.

The WASP ensemble aspect is not really to do with the structure of WASP itself, but the method through which the final WASP ensemble is generated. In the prior ensemble, many model coefficients are varied independently. These simulations are then forced historically and an posterior ensemble is generated, where each simulation accepted into the posterior agrees with historic observations. In the posterior ensemble the coefficient values are dependent upon one another: the combination of coefficient values must produce a historically consistent simulation.

The historic constraints on atmospheric carbon are much narrower than the historic constraints on land carbon and ocean carbon, and therefore the posterior ensemble contains simulations whose combined land and ocean carbon cycle responses produce a very narrow atmospheric carbon history. This compensation in the posterior ensemble may carry through when the ensemble is forced with idealised scenarios, since the WASP model coefficient values are dependent upon one another in the posterior ensemble.

For ZECMIP, we see that the coefficient of variation for the landborne fraction is much larger than that for the atmosphere and ocean. Hence there are some aspects of the land response that are being compensated for by the ocean response. This compensation will also apply to the WASP ensemble: If land carbon fraction were high, then the atmosphere fraction would be lower, in turn making the ocean fraction lower. Therefore, the atmosphere fraction is reduced by less than initially expected from the high land carbon fraction – while a high land carbon fraction takes

directly from the air, this results in reduced ocean fraction which compensates to reduce the impact on atmospheric fraction. This effect is seen in both the WASP and ZECMIP ensembles.

line 365 - "carbon feedbacks" - I don't see that carbon feedbacks are discussed at all in this paper.

Carbon feedbacks are connected to the carbon inventory changes, but agreed in this study we have not explicitly diagnosed the carbon feedbacks (we have done this in Arora et al. (2020) doi:10.5194/bg-2019-473). Rephrased to carbon responses.

line 386 - no discussion of carbon or the range of carbon uptake? The carbon contribution is only minimally smaller than the thermal contribution so warrants further discussion.

We agree with this concern. We have added text referring to the spread in the land carbon responses, which connect to the effect of nutrient limitations on land carbon uptake. It is known that those CMIP6 models with a land nitrogen cycle have smaller carbon feedbacks (Arora et al., 2020). This reduction in the land carbon response is because vegetation growth is limited by nitrogen availability – and this is also visible in ZECMIP results where the models with a land nitrogen cycle have a lower mean land fraction of 0.31 compared to 0.41 in the 3 models without a nitrogen-cycle (here CanESM, CNRM and GFDL). These models are already identified in the paper as having higher land-fraction, but we will add brief discussion of this response to the process-inclusion of land nitrogen cycle in the manuscript.

L407 i) for the carbon contribution, some models (CESM2, CNRM-ESM2) have the land sink strengthening in time and always dominating over the ocean, while other models (ACCESS-ESM1.5, UKESM1) have the land sink saturating due to nitrogen limitation and the ocean sink eventually dominating — the former response leads to a strengthening in the magnitude of the carbon contribution and acts to give a negative ZEC;

line 398-402 - is there a way to visualize the tradeoffs described in this paragraph?

This information is already conveyed in Figure 4 where the magnitude of the different lines corresponds to the strength of that process. For example, a strong thermal amplification is represented by the red line being greater than 1, while a strong carbon cycle is represented by a large decrease in the green line. The product of these factors then defines the geometric ZEC.

line 406 - how are carbon climate feedbacks being assessed in this paper?

It is the climate feedbacks that are assessed and diagnosed in Figure 7 red line and reported in Table 1c. We do not separately diagnose the carbon-climate feedbacks in this study, but they are diagnosed in Arora et al. (2020) doi:10.5194/bg-2019-473.

**Technical Corrections**

Egn 7, Egn 19, Egn 20 - labels for terms are offset

Agreed, aligned.

**References**

B. M. Sanderson, V. Brovkin, R. Fisher, D. Hohn, T. Ilyina, C. Jones, T. Koenigk, C. Koven, H. Li, D. Lawrence, P. Lawrence, S. Liddicoat, A. Macdougall, N. Mengis, Z. Nicholls, E. O'Rourke, A. Ro-manou, M. Sandstad, J. Schwinger, R. Seferian, L. Sentman, I. Simpson, C. Smith, N.

Steinert, A. Swann, J. Tjiputra, and T. Ziehn. flat10mip: An emissions-driven experiment to diagnose the climate response to positive, zero, and negative co2 emissions. EGUsphere, 2024:1–39, 2024. https://doi.org/10.5194/egusphere-2024-3356

S. Palazzo Corner, M. Siegert, P. Ceppi, B. Fox-Kemper, T. L. Fr'ölicher, A. Gallego-Sala, J. Haigh, G. C. Hegerl, C. D. Jones, R. Knutti, C. D. Koven, A. H. MacDougall, M. Meinshausen, Z. Nicholls, J. B. Sall ee, B. M. Sanderson, R. S ef erian, M. Turetsky, R. G. Williams, S. Zaehle, and J. Rogelj. The zero emissions commitment and climate stabilization. Frontiers in Science, Volume 1 - 2023, 2023.

**Both above references are cited.**

Thank you for the detailed points raised, which have been helpful in making the manuscript clearer and more explicit as to the benefits of the geometric ZEC and the normalised framework.

---

## Author Response (AR2)

**Response to the second set of reviews**

One referee had no further comments and recommended acceptance, while another referee with a second reading had further minor comments to address. We thank the referees for their positive recommendations and attention to detail.

**Comments to address**

In general the authors have answered many of my questions within the response to reviewers document but have not clarified the main text in response. The authors make several useful arguments in the response to reviewers that should be added to the main text prior to publication.

First example, the authors have in the response to reviewers that "The framework is designed to provide a quantitative measure of the different drivers of the ZEC. Without a quantitative measure, one is left making qualitative comparison of thermal and carbon effects when those variables are measured in different ways. Our framework provides a formal way of comparing the relative importance of each driver." This would be very helpful context for readers early on in the MS.

We thank the referee for the additional comments and by recommending that we are more explicit in making these connections.

**Added in Abstract, L5-7:**

In order to understand these different climate responses, a normalised framework is introduced that quantifies the relative importance of carbon, radiative and thermal drivers of the ZEC.

**Expanded in the Introduction, L39-42:**

A framework is introduced that formally compares the relative importance of these

thermal, radiative and carbon drivers for the ZEC (Section 2). Without a quantitative measure, only a qualitative comparison of thermal and carbon effects can be made, which is complicated by those variables being measured in different ways. These

drivers for the ZEC are interpreted

**Expanded in the Conclusions, L403-404:**

In order to gain mechanistic insight as to the controls of the ZEC, a normalised framework is introduced that formally compares the relative importance of thermal, radiative and carbon drivers for the ZEC.

Modified in the Conclusions, L472-474:

In summary, our normalised framework provides a formal comparison of the different thermal, radiative and carbon contributions to the ZEC, and so provides mechanistic insight as to why there are different temperature responses from Earth system models after carbon emissions cease.

Second example, "For example, for the ZECMIP diagnostics for the geometric ZEC, we find that..." (three points follow)

The framework is designed to provide a quantitative measure of the different drivers of the ZEC. Without a quantitative measure, one is left making qualitative comparison of thermal and carbon effects when those variables are measured in different ways. Our framework provides a formal way of comparing the relative importance of each driver.

For example, for the ZECMIP diagnostics for the geometric ZEC, we find that

- (i) The intermodal spread in the geometric ZEC is primarily controlled by the intermodal spreads in the normalised thermal contribution and normalised atmospheric carbon concentration, rather than that of the normalised radiative forcing dependence on atmospheric CO2 (Table 1b):
- (ii) The intermodal spread of the normalised contribution to the warming dependence on radiative forcing is mainly determined in ZECMIP by the intermodal spread in the fraction of radiative forcing returned to space rather than that of the inverse climate feedback (Table 1c);
- (iii) The intermodal spread of the airborne fraction is mainly determined by the intermodal spread of the landborne fraction, rather than the oceanborne fraction (Table 1c).

I don't see these three conclusions clearly stated in the manuscript. This seems like relevant context for presenting this new metric of ZEC - please include it in the main text.

These points were separately explained in the main manuscript, but we have modified the Conclusions to explicitly reiterate these points:

**Conclusions, L428-437:**

Applying our normalised framework to the ZECMIP diagnostics reveals that relative importance of the different drivers for the inter-model spread of the geometric ZEC: (i) The inter-model spread in the geometric ZEC is primarily controlled by the intermodal spreads in the normalised thermal contribution and normalised atmospheric carbon concentration, rather than that of the normalised radiative forcing dependence on atmospheric CO2 (Table 1b); (ii) The inter-model spread of the normalised contribution to the warming dependence on radiative forcing is mainly determined in ZECMIP by the intermodal spread in the fraction of radiative forcing returned to space rather than that of the inverse climate feedback (Table 1c); (iii) The inter-model spread of the airborne fraction is mainly determined by the intermodal spread of the landborne fraction, rather than the oceanborne fraction (Table~1c).

"We find that the model responses separate into different classes. For the carbon response, the land carbon sink either continues to increase in time or saturates. These different responses appear to be linked to whether there is a nutrient cycle that can inhibit the ability of the land to take up unlimited carbon."

The authors make this point in the abstract and at the end of the conclusions, however it would be helpful to connect this simple summary with the discussion of land sink saturation (implied as due to nutrients ~ line 255).

We have added new text to discuss the land carbon response, see below.

**Third example**

"For example for the inter-model spread in the climate responses, the framework reveals..." followed by three points.

I can see that these points are made in the text, but not as directly as they are here. I think the MS would benefit from directly stating these findings.

As above we have included these points in the Conclusions, L428-437.

Applying our normalised framework to the ZECMIP diagnostics reveals that relative importance of the different drivers for the intermodel spread of the geometric ZEC: (i) The intermodel spread in the geometric ZEC is primarily controlled by the intermodal spreads in the normalised thermal contribution and normalised atmospheric carbon concentration, rather than that of the normalised radiative forcing dependence on atmospheric CO2 (Table 1b); (ii) The intermodel spread of the normalised contribution to the warming dependence on radiative forcing is mainly determined in ZECMIP by the intermodal spread in the fraction of radiative forcing returned to space rather than that of the inverse climate feedback (Table 1c); (iii) The intermodel spread of the airborne fraction is mainly determined by the intermodal spread of the landborne fraction, rather than the oceanborne fraction (Table~1c).

With regards the quantification of our analyses, we have added new comparisons linked to Table A1 providing estimates of the relative magnitude of the different contributions to the variance of ZEC, the carbon contribution to the ZEC and the thermal contribution to the ZEC:

**New paragraph, L236-241**

The normalised contributions to ZEC from the thermal, radiative and carbon responses can also be analysed to quantify the contribution of each term to the spread across models. By varying just one model input term in Table A1, the thermal and carbon terms explain 58% and 40% respectively of the variance in ZEC, whereas the radiative term explains only 2% of the variance. This analysis confirms that both the model spread in thermal response and the model spread in carbon sink both contribute significantly to

the spread in ZEC, and both remain high priority research areas to understand in order to reduce uncertainties in ZEC.

These competing carbon and thermal contributions for the ZEC are next addressed in more detail in terms of their own dependencies.

**New paragraph, L274-279**

In common with the analysis of Jones and Friedlingstein (2020), by varying just one model input term at a time in Table A1, the land carbon sink is again found to dominate the spread in the carbon sink contribution to ZEC, accounting for 78% of the variance in the carbon sink compared with the ocean sink explaining 22% of the variance at 50 years after net zero. The magnitude of land and ocean sinks are similar on this timescale, but the model spread is greater for the land sink. On longer timescales beyond a century, we expect the land carbon sink to saturate more rapidly and the ocean carbon sink to play a progressively more important role.

**Line 306-308**

In addition, by varying just one model input at a time in Table A1, the fraction of radiative forcing returning to space explains 67% of the variance in the thermal contribution to the ZEC compared with 33% from the variance in the inverse climate feedback parameter.

**Evidence for nutrient limitation:**

While nutrient limitation seems like a very plausible hypothesis for saturating land carbon sinks I don't see any analysis or quantification of evidence for that presented in this MS. Please provide evidence or make it clear that this statement is a hypothesis that needs to be tested. For example CESM2 has an increasing sink but includes interactive nutrient cycling. Also I don't see evidence that it is nitrogen specifically causing the nutrient limitation (ACCESS-ESM1.5 also has phosphorus represented). If the authors argue that this is covered already by Aurora et al. 2020 they need to make that argument in the MS more clearly.

The statements about nutrient limitation are more than a hypothesis and are in accord with two other published studies by Zaehle et al. (2015) and Ziehn et al. (2021), both demonstrating the limitation of the land carbon sink through nutrient availability. The models with the smallest landborne fraction are those that have a nutrient limitation. For CESM2, there may still be an increase in the land carbon sink, but the increase in

the land carbon sink is less than it would be without nutrient limitation. Models usually represent nutrient limitation in terms of nitrogen limitation (the exception is ACCESS that has phosporus limitation). So the separation between models is more typically represented by whether nitrogen limitation included.

Added new text on the land carbon sink, L260-270

These different relative strengths of the land and ocean carbon sinks are likely due to structural differences in the land carbon model which contributes a much greater model spread than the ocean response (Jones and Friedlingstein., 2020). The three models with higher landborne fractions may be overestimating the possible land carbon sink as they neglect the role of nutrient limitations, while the other models include land nitrogen cycling and limitation of carbon allocation. Models without an explicit terrestrial nitrogen cycle can project unrealistic land carbon sinks which could not be supported by available nutrients (Zaehle et al., 2015). In particular, Ziehn et al. (2021) showed explicitly for the ACCESS model, the important role of nutrients – both nitrogen and phosphorus - in reducing land carbon sinks. Arora et al. (2019) showed this inclusion or absence of nutrient limitation to be the largest systematic difference in the carbon response of CMIP6 Earth system models, with a distinct split in the land carbon response to climate and CO2 between models with and without a nitrogen cycle. Consequently, the inclusion of nitrogen limitation on land acts to reduce the increase in the projected land carbon sink and so acts to increase the resulting ZEC.

**Two new references added:**

Zaehle, S., Jones, C. D., Houlton, B., Lamarque, J.-F., and Robertson, E.: Nitrogen availability reduces CMIP5 projections of twenty-first-century land carbon uptake, Journal of Climate, 28, 2494–2511, 2015.

Ziehn, T., Wang, Y., and Huang, Y.: Land carbon-concentration and carbon-climate feedbacks are significantly reduced by nitrogen and phosphorus limitation, Environmental Research Letters, 16, 074 043, 2021.

**Also added in Conclusions**

L441 while other models (ACCESS-ESM1.5, UKESM1) have the land sink eventually saturating due to nitrogen limitation and the ocean sink dominating

discussion of WASP: "There are two reasons for this "coefficient of variation being larger for the landborne and oceanborne fractions than the airborne fraction", one to do with the WASP ensemble and another to do with the system itself..." (and following response) Please add at least a sentence or two making this clarification in the main text.

Agreed. We have added the text L385-387:

This reduced spread of the airborne fraction in WASP is also partly due to how the WASP ensembles are constructed with historic constraints on atmospheric carbon being much narrower than the historic constraints for land and ocean carbon.

I don't find the term "geometric ZEC" intuitive but I am not opposed to this label. We think that this term is mathematically accurate and avoids the ambiguity of using term of normalised ZEC (as raised previously by the referee), where different choices are possible.

As a note the MS uses intermodel, inter-model, and in the response to reviewers intermodal. Please make consistent.

Changed to inter-model.

line 225 - it would be helpful to add "which are discussed in further detail below" or something similar

Agreed, added on L224

; these contributions are discussed in more detail in the next subsections.

comment about original line 398-402

The text has been modified more than is represented by the highlighted version. Either way I think it would be helpful to add the references to where the evidence for these statements is found in the MS (i.e. Fig 4).

The new changes are highlighted in the pdf.

Added the extra figure references to support the points made:

```
L4443 (Fig. 4, green line; Fig. 5)
L446 (Fig. 4, red line; Fig. 6);
L448 (Fig. 4),
```

Figure 9. I understand that the authors don't wish to do further analysis of additional datasets, but I don't see why this couldn't in principle be done for the existing flat10MIP ESM runs.

In principle the analyses can be repeated for the flat 10 scenario, but those analyses would delay the manuscript. The relative importance of the thermal and carbon

contributions is unlikely to be determined by the scenario. For example, we do provide an analysis for 1pctCo2 and flat10 for WASP integrations in Figures 9 and 10, and Tables S2 and S3, and the responses only differ in detail.

Added in Conclusions, L463-465

To partly address the above caveats, diagnostics of a large ensemble of an efficient Earth system model (WASP) are also examined for 2 different scenarios (1pctCO2 and flat10). Their ensemble median responses for the ZEC and its contributions are broadly similar for both scenarios and to those of ZECMIP.

L494 We have included thanks for the constructive comments that have strengthened the study.